# Gradient in cytoplasmic pressure in germline cells controls overlying epithelial cell morphogenesis

Laurie-Anne Lamiré[1]©, Pascale Milani[1]©, Gaël Runel[1], Annamaria Kiss[2], Leticia Arias[1], Blandine Vergier[1], Stève de Bossoreille[1], Pradeep Das[2], David Cluet[1]*, Arezki Boudaoud[2]*, Muriel Grammont[1]*

1 Laboratoire de Biologie et de Modélisation de la Cellule, Univ Lyon, ENS de Lyon, UCB Lyon 1, CNRS, Lyon, France, 2 Reproduction et Développement des Plantes, Univ Lyon, ENS de Lyon, UCB Lyon 1, CNRS, INRAE, Lyon, France

© These authors contributed equally to this work.
* david.cluet@ens-lyon.fr (DC); arezki.boudaoud@ens-lyon.fr (AB); muriel.grammont@ens-lyon.fr (MG)

**Data Availability Statement:** All relevant data are within the paper and its Supporting Information files.

## Abstract

It is unknown how growth in one tissue impacts morphogenesis in a neighboring tissue. To address this, we used the *Drosophila* ovarian follicle, in which a cluster of 15 nurse cells and a posteriorly located oocyte are surrounded by a layer of epithelial cells. It is known that as the nurse cells grow, the overlying epithelial cells flatten in a wave that begins in the anterior. Here, we demonstrate that an anterior to posterior gradient of decreasing cytoplasmic pressure is present across the nurse cells and that this gradient acts through TGFβ to control both the triggering and the progression of the wave of epithelial cell flattening. Our data indicate that intrinsic nurse cell growth is important to control proper nurse cell pressure. Finally, we reveal that nurse cell pressure and subsequent TGFβ activity in the stretched cells combine to increase follicle elongation in the anterior, which is crucial for allowing nurse cell growth and pressure control. More generally, our results reveal that during development, inner cytoplasmic pressure in individual cells has an important role in shaping their neighbors.

## Introduction

Epithelial cells collectively adopt specific shapes when building organs. Over the last four decades, it has been demonstrated that cell and organ morphogenesis depend on the integration of extrinsic and intrinsic biological cues (endocrine, paracrine, or autocrine signaling; cell-cell or cell-extracellular matrix adhesion; actin filament and microtubule organization, etc.). Internal forces generated by the activity of the actomyosin network have been shown to be crucial for cell shape, by acting on adherens junction (AJ) and/or on cell-extracellular matrix adhesive complexes [1,2,11–18,3–10]. Such studies shed light on the importance of considering the mechanical properties of cells, such as cortical tension, to understand morphogenesis [19–22]. However, cell shape not only is imposed by intrinsic forces but also depends on the local environment, which comprises other types of cells and extracellular matrix. The

**Funding:** This work received supported by the Agence Nationale de la recherche (Blanc 12-SVSE-0023-01, MechInMorph to AB and MG) (https://anr.fr), the Centre National pour la Recherche Scientifique (https://www.cnrs.fr), and the Ecole Normale Supérieure of Lyon (http://www.ens-lyon.fr). The funders had no role in study design, data collection and analysis, decision to publish, or preparation of the manuscript.

**Competing interests:** The authors have declared that no competing interests exist.

**Abbreviations:** A.U., arbitrary units; A/P, antero-posterior; AFM, atomic force microscope; AJ, adherens junction; ANC, anteriormost NC; BM, basement membrane; dic, dicephalic; Ecad, Ecadherin; ERC, entrance RC; ES9, early S9; Eya, eye-absent expression; GFP, green fluorescent protein; kel, kelch; LS9, late S9; MARS pipeline, Multi-Angle image acquisition, three-dimensional Reconstruction, and cell Segmentation; MBFC, main body follicular cell; MS9, mid S9; NC, nurse cell; pMad, phosphorylated form of Mad; PNC, posteriormost NC; pSqh, phosphorylated sqh; RC, ring canal; RFP, red fluorescent protein; S, stage; sqh, spaghetti squash; StC, stretched cell; TkvA, constitutively active form Tkv; WT, wild type; Yki, Yorkies.

mechanical properties of these biological elements may also impact and deform epithelial cells. The implication in epithelial morphogenesis of such extrinsic forces is far less investigated and known, compared to intrinsic forces [23,24].

With its simple structure and its well-characterized pattern of development, the *Drosophila* ovarian follicle is a valuable model to study epithelial cell morphogenesis. A follicle consists of an inner cyst of 16 germinal cells (15 nurse cells [NCs] and one posteriorly localized oocyte) surrounded by a monolayered epithelium of about 800 cells, which are themselves covered by an outer basement membrane (BM) (Fig 1A). Follicle development has been divided in 14 stages (S), with S1 and S14 corresponding to a follicle emerging from the germarium and a mature egg, respectively [25,26]. The follicle is initially small and spherical with a 10-μm diameter. It progressively grows up and elongates before undergoing a major acceleration of growth and elongation at S9. In parallel, the oocyte is gradually enlarged compared to individual NC, owing to the presence of ring canals (RCs) between all the germline cells that serve to transfer NC cytoplasmic contents into the oocyte [27]. The epithelial cells accommodate the germline growth by first proliferating from S1 to S6 to reach about 850 cells and second by increasing in volume and changing in shape at S9 [25,26]. Starting from a cuboidal shape, about 50 cells, called the stretched cells (StCs), flatten above the NCs whereas the others, called the main body follicular cells (MBFCs), become columnar above the oocyte. StC flattening occurs progressively from the anterior to the middle of the follicle and can be monitored by following the increase of the apical surface (Fig 1B). This morphogenetic process depends on the activation of specific genes in the StC, such as those involved in the TGFβ pathway and the *tao* gene, which control the sterotyped disassembly of the subapical AJs (Fig 1B') and the lateral adhesion complexes, respectively [28–30]. Cell flattening also comes with changes in BM mechanical properties and interactions, which are both controlled by the TGFβ pathway [31]. Finally, Kolahi and colleagues showed that StC flattening depends on the growth of the germline cells that controls the degree of flattening [32]. How this external force influences StC flattening has never been investigated.

In this study, we establish that a gradient of mechanical properties exists in the NC compartment along the antero-posterior (A/P) axis and that this gradient is responsible for the progressive change of shape of the StC. We show that the differences in NC mechanical properties correspond to differences in cytoplasmic pressure, which is the force that counters cortical tension. We present evidence that pressure level is modulated by intrinsic NC growth and controls TGFβ expression dynamics. We reveal that NC pressure leads to inhomogeneous follicle expansion, with a bias toward the anterior and that this expansion requires StC flattening and BM softening. We show that anterior expansion is crucial for the maintenance of NC integrity while their inner pressure builds up. Whereas virtually no study has paid attention to cytoplasmic pressure as a physical component of the cell, our data bring to light its role in imposing shape in epithelial cells and tissues.

## Results

### The wave of StC flattening does not depend solely on TGFβ activity

StC flattening occurs progressively from anterior to posterior and depends on TGFβ signaling [29]. To determine whether the wave of flattening depends on a wave of TGFβ activity, we quantified the expression of the phosphorylated form of Mad (pMad) from S8 to S10. pMad is mainly detected in epithelial cells above the anterior and central NC with the highest levels in the anterior cells. Only a weak expression is detected in the epithelial cells surrounding the posterior NC. As StC flattening progresses posteriorly, pMad level increases in all the epithelial cells that surround NC, with a maximum detected in those above the central part. These data

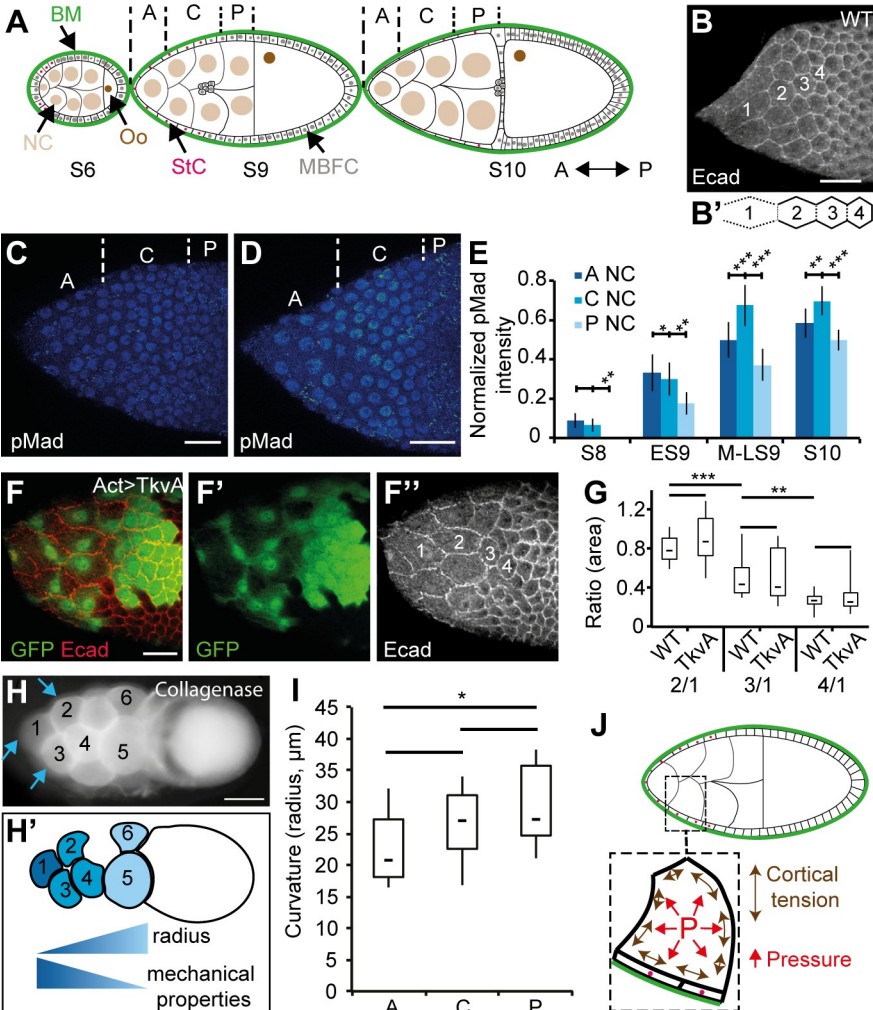

**Fig 1. An A/P gradient of mechanical properties is present in the NCs.** In all figures, anterior is to the left. (A) Schematic representation of three follicles at different developmental stages (S6, S9, and S10) with NCs, Oo, MBFCs, StCs, and BM indicated. From S8 onward, NCs are grouped into three regions for analysis: anterior ("A"), central ("C"), and the four posterior ("P") NCs. (B) AJs (Ecad) remodeling and increase of apical surface occur in the StC of an S9 follicle. Both variables are marker of flattening (see S1 Movie). Four cells are labeled from anterior ("1") to posterior ("4") to highlight changes in apical surface. (B') Schematic representation of AJ remodeling, with dotted lines representing remodeled AJ and solid lines representing intact AJ. The AJ disassembly consists of first remodeling the vertices on the same row and the AJ perpendicular to the A/P axis; second, elongating the AJ parallel to the AP axis; and third, disassembling those AJ (Fig 1B'). (C, D) pMad expression in early (C) and mid (D) WT S9 follicles. The anterior ("A"), central ("C"), and posterior ("P") regions of the NC are indicated. (E) Quantification of pMad in follicular cells during StC flattening ($n > 15$ nuclei per condition). (F) TkvA-expressing StC in an S9 follicle marked by GFP (F and F'), and with StC labeled in increasing order from anterior to posterior (F"). (G) Box and whisker plot of the ratio of surface area of StC in WT or TkvA-expressing StC at the different positions shown in (B) or (F"), respectively ($n > 23$ cells per plot). (H) A WT S10 follicle after collagenase treatment, showing the NC bulging outward, with the corresponding schematic representation of the mechanical property gradient based on NC membrane curvature (the measured curvature corresponds to the membrane bulging outwards). Blue arrows point to the region of the membrane used to calculate the curvature (H'). (I) Box and whisker plot of the radius of curvature of anterior ("A"), central ("C"), or posterior ("P") NC ($n > 13$ per region) in LS9 and S10 follicles. (J) Schematic representation of an S9 follicle, with a blow up on an NC, its overlying StC, and the BM. In the NC, cortical tension (double-sided arrows, in brown) and cytoplasmic pressure ("P," in red) are represented. Scale bars: 20 μm (B-F) and 50 μm (H). Data for graphs (E), (G), and (I) can be found in the S1 Data file. In box and whisker plots in all figures, boxes extend from 25 to 75 percentile, with a line showing the median value. Whiskers extend to the most extreme values. In all figures with panels displaying levels in fluorescence intensity, a color-coded gradient from blue (low) to red (high) is used. In all figures, error bars indicate s.e.m. In all figures, *, **, and *** correspond to $p < 0.5$, $p < 0.05$, and $p < 0.01$ (*t* test), respectively. AJ, adherens junction; A/P, antero-posterior; BM, basement membrane; Ecad,

Ecadherin; ES9, early S9; GFP, green fluorescent protein; LS9, late S9; MBFC, main body follicular cell; MS9, mid S9; NC, nurse cell; Oo, oocyte; pMad, phosphorylated form of Mad; S, stage; StC, stretched cell; TkvA, constitutively active form of Tkv; WT, wild type.

confirm that TGFβ activity levels vary in the cells covering the NC and that these variations follow the progression of StC flattening (Fig 1C–1E) [33]. To determine the importance of the spatiotemporal pattern of TGFβ for the progressive change of shape, we measured StC apical surface area when expressing a constitutively active form of the TGFβ receptor Thick veins (Tkv), which we refer to as TkvA. Our data show that the TkvA-expressing StC still flatten like a wave from anterior to posterior, demonstrating that cell flattening remains progressive in the absence of graded TGFβ activity (Fig 1F and 1G). We conclude that another variable, independent from the TGFβ pathway, must therefore control the wave of flattening.

## An A/P gradient of mechanical properties is present in the NC compartment

We previously observed that a collagenase treatment on follicles leads to the bulging of the NC toward the outside, when the constraint imposed by the BM is removed (Fig 1H) [31]. This also shows that StC are insufficient to constrain NC under collagenase treatment. Furthermore, this experimental condition facilitates the assessment of NC mechanical properties. Cell mechanical properties are made up of mainly two components: cortical tension, controlled by the cortical actomyosin network and by cell-cell adhesion, and the opposed cytoplasmic pressure, which may vary with solute and water fluxes, with differences in volume or growth rate, and with organization of cytoplasmic structural elements, such as microtubules and noncortical actin microfilaments (Fig 1J). We reasoned that the extent of curvature of the outward-facing NC membrane (the radius of the circle fitting this membrane) may reflect NC mechanics. Indeed, the radius of curvature (R) of a spherical interface is proportional to the ratio of its tension (γ) and inversely proportional to the difference in pressure (ΔP) between the two sides of the interface: $R = 2\gamma/\Delta P$, which generalizes to nonspherical interfaces by replacing $2/R$ by the mean curvature of the interface [34,35]. Because of limited resolution along the optical axis of the microscope, we approximated NC mean curvature by $2/R$, $R$ being a radius measured in 2D optical sections along the A/P axis. We found a gradient of curvature with smaller radii at the anterior in late S9 (LS9) and S10 follicles (Fig 1H and 1I), indicating that NC mechanical properties vary along the A/P axis, with tension increasing and/or pressure decreasing from anterior to posterior.

## The A/P gradient of NC mechanical properties is consistent with differences in cortical tension at LS9

To test whether the observed variations in NC curvature from anterior to posterior could be due to increasing NC surface tension, we considered surface tension of the NC during StC flattening in follicles with intact BM. The tension (γ) of a cell-cell interface is increased by contractility of the actomyosin cortex ($\gamma_{am}$) and reduced by cell-cell adhesion ($\gamma_{ad}$) and can be stated as $\gamma = \gamma_{am} - \gamma_{ad}$ [36–38]. Accordingly, we analyzed cortical actomyosin dynamics through the expression and activity of the regulatory chain of the nonmuscular Myosin II (encoded by *spaghetti squash* [*sqh*]) and cell-cell adhesion through the expression of Ecadherin (Ecad), in live and fixed follicles from S8 to S10 (Figs 2A and S1A–S1C). Ecad and Sqh (both the nonphosphorylated and the phosphorylated [pSqh] forms) are strongly detected at the cortex of the oocyte from S8 to S10. Ecad expression is weak in the NC at S8 and increases during

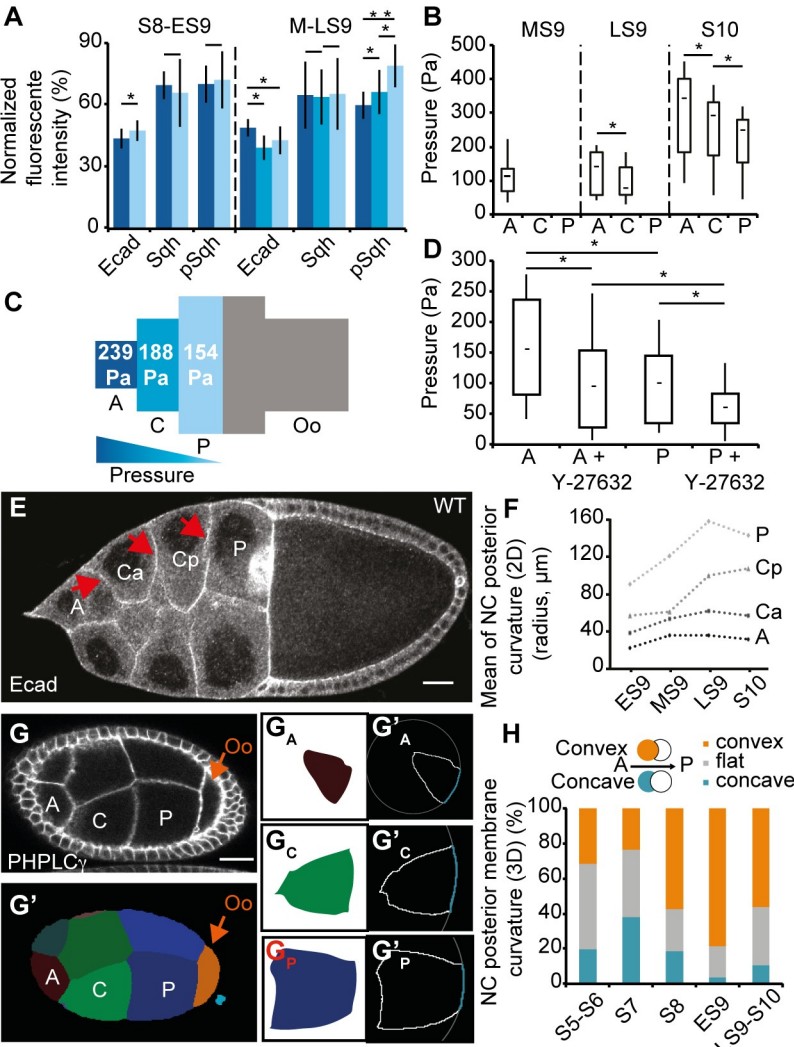

**Fig 2. AFM measurements and NC geometry reveal an A/P gradient of cytoplasmic pressure.** (A) Percentage of Ecad::GFP (live follicles), Sqh::RFP (live follicles), and pSqh (fixed follicles) expression at the interface between anterior and central (dark blue) NC, or between central abutting the anterior NC and central abutting the posterior NC (cyan, only at MS9 and LS9), or between central and posterior (light blue) NC, relative to the expression between posterior NC and the Oo (*n* > 15 per stage). (B) Box and whisker plot of cytoplasmic pressure deduced from AFM in anterior ("A"), central ("C"), and posterior ("P") NC, from MS9 to S10 WT follicles (*n* = 10 for MS9 and LS9 and *n* = 6 for S10). (C) Schematic representation of mean pressure deduced from AFM in each region of WT S10 follicles, with a color-coded gradient from blue (high) to green (low) (*n* > 12 per position). (D) Box and whisker plot of cytoplasmic pressure deduced from AFM in anterior ("A") and posterior ("P") NC, at MS9 WT follicles before or after treatment with ROCK inhibitor (Y-27632) (*n* = 13 per bar). (E) An LS9 follicle showing the curvature of the membranes (arrows) between the anterior ("A") and a central NC ("Ca"), between two central NC ("Ca" and "Cp"), or between a central ("Cp") and a posterior ("P") NC. "Ca" and "Cp" refer to a central NC in contact with an anterior NC or a posterior NC, respectively. (F) Mean of the NC posterior membrane curvatures (as labeled in E) at either ES9 (*n* = 98 cells), MS9 (42 cells), LS9 (25 cells), or S10 (15 cells). (G) An S8 follicle and the corresponding section from a 3D segmentation of the same (G') with NC labeled from anterior to posterior. $G_A$, $G_C$, and $G_P$ correspond to slices showing the largest area for each labeled NC. $G'_A$, $G'_C$, and $G'_P$ show the outline (solid white line) of each cell, as well as the circle (dotted white line) fitted to the posterior membrane (blue line). (H) Percentage of convex (orange), flat (gray), and concave (blue) NC posterior membrane curvatures at different follicular stages (2–3 follicles per stage were reconstructed, *n* > 70). Scale bars: 20 µm. Data for graphs (A), (B), (C), (D), (F), and (H) can be found in the S1 Data file. A/P, antero-posterior; AFM, atomic force microscope; Ecad, Ecadherin; ES9, early S9; GFP, green fluorescent protein; LS9, late S9; MS9, mid S9; NC, nurse cell; Oo, oocyte; pSqh, phosphorylated sqh; RFP, red fluorescent protein; S, stage; sqh, spaghetti squash; WT, wild type.

S9. Sqh expression is undetectable at S8 but starts to be visible at S9. In S9 follicles, pSqh is also detected at a low level at the NC cortex. To quantify and compare Ecad, Sqh and pSqh expressions along the A/P axis, expression levels at the NC membranes along the A/P axis were normalized for each follicle with that of the oocyte cortex. The anterior NC usually express a higher Ecad level than the central NC from mid S9 (MS9) to S10, indicating higher adhesion in the anterior, negatively influencing tension there. Sqh levels remain constant at all NC interfaces, regardless of position, throughout these stages, though the active form of Sqh does show an inverse gradient of expression with higher levels in central and posterior cells compared to anterior cells at the end of S9, indicating lower contractility at the anterior at this stage. Altogether, these observations suggest that a postero-anterior gradient of cortical tension is present from MS9 to LS9.

## The A/P gradient of NC mechanical properties corresponds to differences in cytoplasmic pressure

To test whether the observed variations in NC curvature from anterior to posterior could be due to decreasing NC cytoplasmic pressure, we used an atomic force microscope (AFM), a standard approach to noninvasively measure the stiffness of biological samples [39–45]. The apparent stiffness quantifies how easily a body is deformed when a force is applied to it; a high value corresponds to a stiff material. We designed a bespoke AFM-based protocol with deep sample indentation so as to probe NC internal pressure below the BM and the flattened StC (see Materials and methods, S1D–S1F Fig). From the force curves, we first quantified the apparent stiffness along the follicle (S1G Fig). In parallel, we also quantified the curvature of the follicle surface at the measurement points. With the values of indentation depth used, we extracted pressure using stiffness and curvature values (S1H Fig) (see Materials and methods), as previously described [45]. We observed that inner pressure of anterior NC increases by a factor of 3 during StC flattening and that an A/P gradient of pressure is present, with the highest pressure being in the most anterior probed area (Figs 2B and 2C and S2A). Although NC pressure displays a broad range of values, between 100 and 450 Pa, we observe that it varies by a factor of 1.2- to 1.5-fold between different regions of a single follicle (S2A Fig), whereas variations within each probed region were negligible (S2B and S2C Fig). We note that the values of pressure measured are comparable to those obtained in mitotic cells [46], chick embryo blastula [47], and mouse blastocyst lumen [48].

Although we chose an indentation depth that is more sensitive to NC cytoplasmic pressure, it is possible that these measurements are also sensitive to NC cortical tension. In order to assess this, we performed measurements before and after the addition of Latrunculin B, which disrupts actin cytoskeleton, or of the pharmacological ROCK inhibitor (Y-27632), which has been shown to inhibit Myosin activity [49,50]. We observed that a mild decrease in pressure is observed after incubation in Latrunculin B (S2D Fig) and that blocking Myosin II decreases pressure by a factor of 1.6 for both anterior and posterior NC (Fig 2D). Consequently, the ratio in stiffness between anterior and posterior NC is preserved after ROCK inhibitor treatment, demonstrating that these differences are mostly due to cytoplasmic pressure and not to cortical tension. In addition, we performed different osmotic treatments and always observed a sharp decrease in stiffness values, demonstrating that AFM measurements are responsive to osmotic changes (S2E–S2G Fig).

Finally, it is known that follicles are surrounded by a stiff BM [31,51]. To determine the importance of the BM in NC pressure, we also performed AFM measurements on collagenase-treated follicles. In such follicles, the A/P pressure gradient is still detected, although the absolute values of pressure are lower than in untreated follicles, possibly because of NC expansion

(S2H Fig). Overall, the data are in favor of the presence of a cytoplasmic pressure gradient within the NC compartment at LS9 and at S10.

Since curvature of a soft interface is physically ascribed to differences in hydrodynamic pressure between the two sides of this interface [34,35], we predicted from the gradient in pressure that the interfaces between NC should be curved toward the posterior in normal conditions. To assess this, we examined NC membrane shape and quantified the curvatures of NC membranes in fixed tissues throughout S9 (Figs 2E, 2F, S3A and S3B). This analysis shows that NC membranes bulge toward the posterior at the same time as StC are flattening and that a gradient of curvature is observed from anterior to posterior NC (those abutting the oocyte), with the anterior NC being the most curved. We also analyzed membrane curvature in follicles treated with the ROCK inhibitor and found that most of the membranes still bulge toward the posterior. This does not support a role for cortical tension in generating this gradient of curvature (S3C and S3D Fig). Finally, to obtain more precise data on membrane curvature, we generated 3D reconstructions of live follicles (S1 Movie). To this end, we used the MARS pipeline (Multi-Angle image acquisition, three-dimensional Reconstruction and cell Segmentation) [52] (S3E–S3H Fig) and additionally developed an ImageJ macro to automatically measure membrane curvature in individual NC (see Materials and methods, S4A Fig). We observed that posterior NC membranes become progressively convex during S8 and S9, with no significant influence of the position of the membrane along the A/P axis (Figs 2G, 2H, S4B, S4C and S4E). This correlates with anterior NC membranes becoming concave (S4F Fig). At S9, 72% of posterior-facing membranes are convex compared to 3% of anterior-facing membranes, whereas no specific orientation is ever observed for the lateral membranes (S4G Fig), consistent with the 2D measurements. Finally, we also noticed that the anterior membranes of the oocyte first bulge into the posterior NC (convex) up to S7 before switching to being concave during S8 (S4D–S4H Fig, S2 Movie), indicating a higher pressure in the oocyte than in neighboring NC until S7 and a lower pressure in the oocyte from S8 onward. Note that the percentages of convex posterior NC membranes remain similar when RNAi against *sqh* is driven in the germline (S4I Fig). Altogether, our data demonstrate that NC mechanical properties are graded along the A/P axis at S8 and S9 because of pressure decreasing from anterior to posterior, preceding a possible increase in tension at LS9.

## The gradient of cytoplasmic pressure controls the wave of StC flattening

These data led us to hypothesize that the gradient of NC pressure could be responsible for the wave of the StC flattening. To test this, we used the *dicephalic* (*dic*) and *kelch* (*kel*) mutants, which have both been described as specifically affecting germline development [53,54]. *dic* mutation leads to reduced germline growth [32], and *kel* mutations are known to prevent normal cytoplasmic transfer between the NC and the oocyte due to the presence of small RCs [27,54]. In *kel* mutants, the inner diameter of the RCs fails to expand, leading to reduced lumen [55]. Based on these phenotypic descriptions, we expected that *dic* mutation would lead to a decrease of pressure whereas *kel* mutations would yield an increase in pressure. To prove this, we first analyzed NC membrane curvatures and observed that only half of the membranes bulge towards the posterior in *dic* follicles, whereas they are all convex in *kel* follicles (Figs 3A–3C and S6A; for staging of *dic* and *kel* follicles, see Materials and methods and S5 Fig). We obtained similar results for *Pendulin* follicles. Mutations in *Pendulin*, which encodes a member of the Importin-alpha protein family, display small RCs [56]. These observations imply that there is no gradient of NC cytoplasmic pressure in *dic* and that the gradient is more pronounced in *kel* and *pendulin* than in wild-type (WT) follicles. In the presence of the ROCK inhibitor, membranes in *dic* follicles still present convex and concave curvatures whereas in

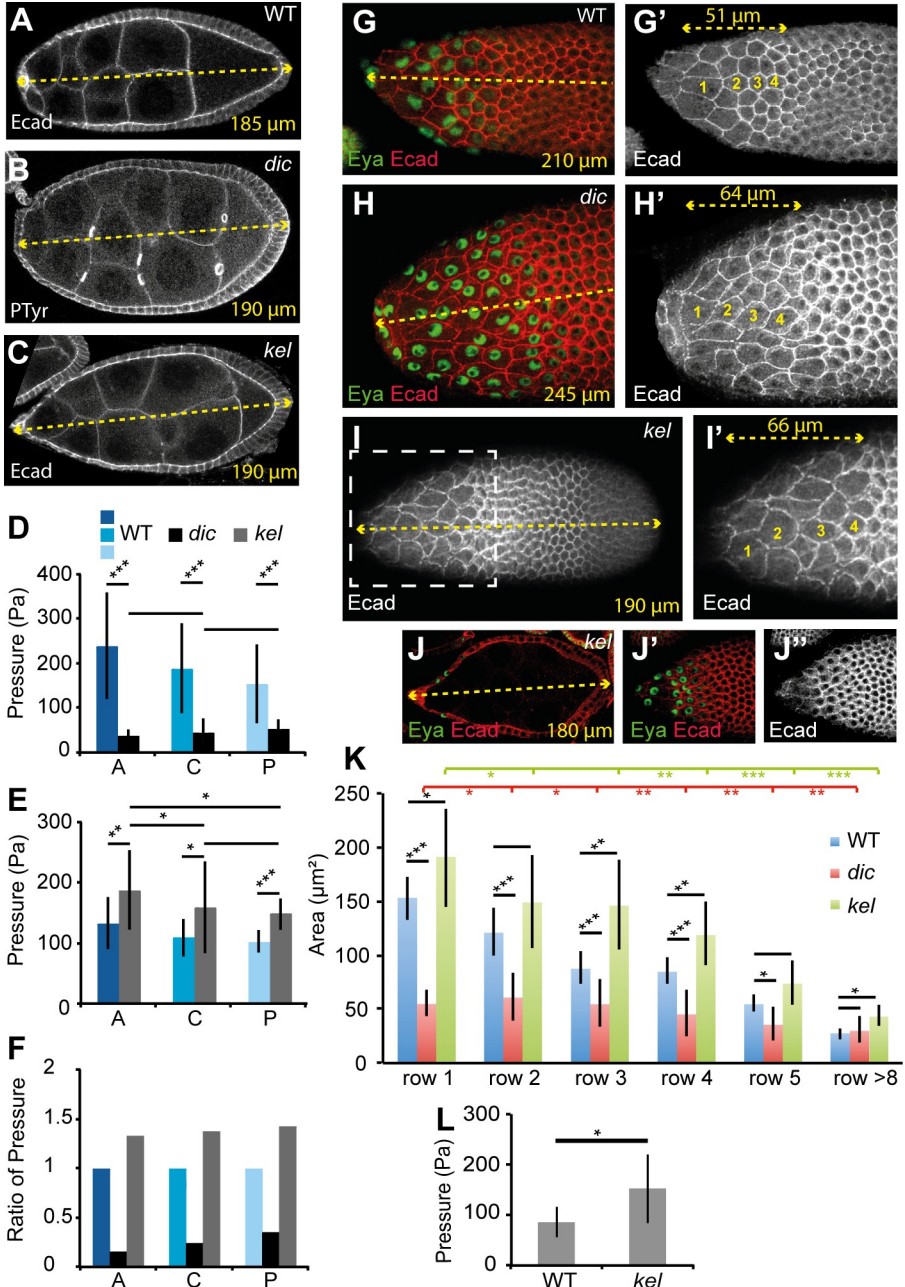

**Fig 3. The gradient of NC pressure controls the timing and the wave of StC flattening.** (A-C) Cross-section of WT (A; S8), *dic* (B; S8), or *kel* (C; ES9) follicles showing NC membrane curvature. The length of the follicle is indicated in yellow. (D) Inner pressure of S10 anterior ("A"), central ("C"), or posterior ("P") NC between *dic* and WT ($n > 12$ for WT, $n = 5$ for *dic*). (E) Inner pressure of anterior, central, or posterior NC between *kel* and WT S10 follicles ($n > 10$ for WT, $n > 7$ for *kel*). Note that WT values in (D) and (E) are specific to each experiment (see Materials and methods). (F) Ratio of pressure in S10 anterior, central, or posterior NC between *dic*, *kel*, and WT. (G-H) StC flattening in mid S9 WT (G) or *dic* (H) follicles. Note that most of the anterior *dic* cells have a similar apical area that the anterior WT cells of row 1 and 2, but for a much longer follicle length (the dashed yellow line corresponds either to the follicle length or to the length of the region of interest). (I, J) *kel* follicle displaying the progressive wave of flattening (I) and a premature StC flattening (J). Surface views (I, I', J', J'') of the follicle or view across the follicle (J); (I') is a magnified view of the box drawn in (I). For (G-I), four cells are numbered in yellow along the A/P axis. Eya is used to visualize StC nuclei. For staging *dic* or *kel* follicles, see Materials and methods. (K) Apical StC surface area by position along the A/P axis in WT (blue), *dic* (red), or *kel* (green) mid S9 follicles ($n > 10$ cells for each row). Statistical comparisons between WT and *dic* or WT and *kel* are indicated in black, between *dic* rows in red and between *kel* rows in green. (L) Inner pressure of WT and *kel* anterior NC in mid S9 follicles ($n = 12$ for WT, $n = 10$ for *kel*). Data for graphs (D), (E), (F),

(K), and (L) can be found in the S1 Data file. A/P, antero-posterior; *dic*, *dicephalic*; Ecad, Ecadherin; ES9, early S9; Eya, eye-absent expression; *kel*, *kelch*; NC, nurse cell; S, stage; StC, stretched cell; WT, wild type.

*kel*, most of the membranes remain convex. In parallel, no significant difference in Ecad or Sqh expression level along the A/P axis has been found between these mutants compared to WT (S6B–S6D Fig), confirming that cytoplasmic pressure, and not cortical tension, is mainly responsible for NC shape in these two mutants (S6A Fig). Second, AFM measurements show that the pressure is much lower in *dic* than in WT in S10 follicles whereas it is higher than WT in *kel* follicles. In *dic* follicles, the pressure gradient is no longer present (Fig 3D and 3F). In *kel* follicles, a difference between anterior and more posteriorly localized NC is still detected, but the values are more variable, as compared to WT (Fig 3E and 3F). These two mutants thus provide valuable mechanical conditions to determine the role of pressure and of the pressure gradient in StC flattening.

We first noticed that the average follicle length with early signs of StC flattening is higher in *dic* follicles and lower in *kel*, compared to WT (Figs 3J and S5). For *dic* follicles, the gradient of flattening is strongly reduced and starts from row 2 (Figs 3G, 3H, 3K and S6E–S6H). In contrast, the flattening remains gradual in *kel* follicles, although it is less pronounced in the central area (Figs 3I–3K and S6E–S6H). In agreement with StC flattening starting prematurely in *kel* follicles, we observed that inner pressure in the anterior NC is higher in MS9 *kel* follicles than in WT (Fig 3L). Together, these data demonstrate that the gradient of pressure controls the wave of StC flattening and that pressure levels dictate the timing of the flattening. It also confirms that NC growth impacts the degree of StC flattening as previously mentioned [32].

## Inner pressure influences the temporal and spatial pattern of TGFβ signaling

We have previously shown that StC flattening depends on TGFβ activity and that premature activation of the TGFβ pathway is sufficient to induce early StC flattening [29]. The data presented above now demonstrate that StC flattening also depends on NC pressure and that contexts with increased pressure, such as in *kel* or *Pen* follicles, trigger flattening prematurely. To examine whether these findings are connected, such that the TGFβ pathway in fact responds to NC pressure, we next monitored and quantified the levels of pMad, a marker of TGFβ activity, in WT and *kel* follicles. When follicle size is used as a benchmark to compare WT and mutant follicles, pMad is present as early as S6–7 in *kel* follicles, whereas it is only detected from S8 onward in WT follicles (Fig 4A and 4B). Additionally, pMad levels were usually 2-fold higher in *kel* follicles than in the WT (Fig 4C–4H and 4M). In both WT and *kel* follicles, the anterior and central areas display higher expression than the posterior area. These data show that high NC pressure is sufficient to induce TGFβ signaling prematurely. In contrast, pMad is detected later in *dic* follicles, which display lower NC pressure than in WT follicles (Fig 4I and 4J). Once expression commences in *dic* mutants, pMad expression levels attain similar levels to the WT but is more uniform (Fig 4K, 4L and 4N). These observations show that pressure levels in the NC are important for the correct temporal and spatial pattern of TGFβ activity.

## High NC cytoplasmic pressure and TGFβ signaling lead to enhanced anterior follicle expansion

Our data show that anterior NCs are more pressurized than posterior NCs and the oocyte when follicles elongate. We reasoned that this might force the growing follicle to expand more anteriorly than posteriorly. To test this, we first used fluorescent beads that adhere to the BM

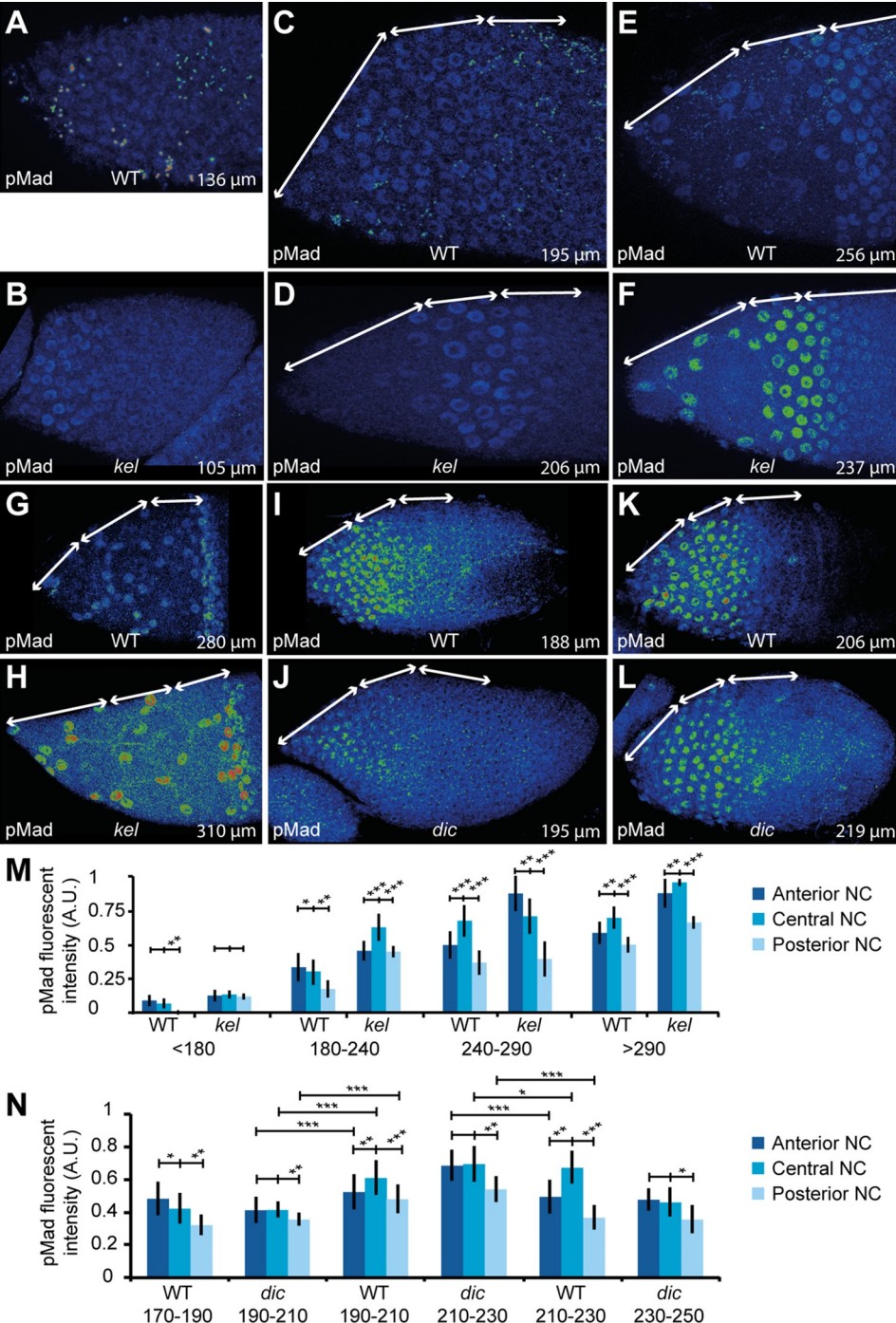

**Fig 4. The pressure gradient controls the timing of the TGFβ pathway.** (A-L) Projections of the sections displaying pMad fluorescence in the StC in WT follicles (A, C, E, G, I, K), *kel* follicles (B, D, F, H), or *dic* follicles (J, L). To allow comparison of expression level between WT and mutants, the images were treated identically for each stage, but differently between stages (enabling comparison between A and B; C and D; E and F; J and K; L and M); (G) and (H) cannot be compared, as intensity levels in (H) were too high to be enhanced as for (G). The double white arrows on each follicle indicate areas above the anterior, central, and posterior NC. The total length of the follicle is indicated. (M, N) Quantification of pMad fluorescence in WT and *kel* follicles (M) or in WT and *dic* follicles (N) in function of follicle length and stage (see S5 Fig) (*n* > 15 nuclei per region). Data for graphs (M) and (N) can be found in the S1 Data file. A.U., arbitrary units; *dic*, *dicephalic*; *kel*, *kelch*; NC, nurse cell; pMad, phosphorylated form of Mad; WT, wild type.

and analyzed their movement during follicle growth (S7A and S7B Fig). From S8 to S10, we observed that beads at the anterior display a greater shift toward the anterior than posterior beads do toward the posterior (*n* = 12) (Figs 5A, 5B, S7C and S7D). Second, we used the *vkg*:: *GFP* line, which expresses one chain of the Collagen IV fused to green fluorescent protein (GFP), resulting in a fluorescent BM. We locally bleached the GFP contained in the BM in order to generate landmark points (*n* = 5) (Fig 5C) and measured elongation of the anterior and posterior BM segments after 2 h. On average, during WT S9, the anterior regions elongate 1.7-fold more than the posterior (Fig 5D), whereas at S7 or S8, no significant difference is detected. In S9 *dic* follicles, where pressure is decreased, the difference between anterior and posterior growth is reduced (*n* = 6) (Fig 5D and S7E Fig). In parallel, we measured absolute follicle elongation and observed that *dic* follicles do not elongate as much as WT follicles, whereas *kel* follicles, where pressure is increased, are more elongated than WT (Figs 5E and S7F–S7H). This suggests that NC pressure biases follicle growth towards the anterior during S9.

We then tested whether this anterior follicle expansion requires StC flattening and BM softening by analyzing follicle elongation when follicular cells constitutively express Dad, which represses TGFβ activity and thus impairs StC flattening and BM softening [29,31]. Our data show that Dad-expressing follicles indeed expand less anteriorly and give rounder follicles and eggs than WT (Figs 5D–5G and S7I–S7J). These data show that anterior follicle and egg elongation requires TGFβ activity in the StC. One explanation for the lack of elongation in Dad-expressing follicles could be from the maintenance of a rigid BM in the anterior, mechanically preventing elongation. If this were true, one would expect that NC inner pressure to be high in Dad-expressing follicles. This is confirmed by AFM measurements that exhibit an increase of pressure by a factor of 1.4 between anterior NC from MS9 WT and Dad-expressing follicles (S7K Fig). Remarkably, we observed that Dad-expressing follicles display NC membrane breakage between the anterior NC at S10, suggesting that NC membranes collapse when pressure increases and anterior expansion is impaired (Fig 5H). From this, we propose that BM softening and StC flattening are important to maintain NC integrity, likely by promoting anterior expansion and maintaining NC pressure under a certain threshold.

## The establishment of the gradient of cytoplasmic pressure depends on intrinsic NC growth

As mentioned above, inner pressure varies as a function of differences in volume/growth rate. To precisely test the role of NC volume, we extracted 3D cell resolution data for each NC from S4 to S10 by using the MARS method (Fig 6A–6F) [52]. Our data show that up to S8, volumes are rather similar between anterior and central NC and that the oocyte is much smaller than any NC (Figs 6G and S8A–S8B). From S8 onward, the volumes of germline cells increase dramatically, especially of the four posterior NC and the oocyte. From S8 to S10, the ratio of the volumes of posterior and anterior NC is 2.4 ± 0.4 (mean ± SD) whereas the ratio of anterior and central NC volumes is only 1.1 ± 0.3. Our data show a gradient of volume between NC is present at these stages, with the anterior NC being the smallest. No significant differences are observed between the central cells. The gradient flattens at S10, except for the four posterior NC. These data demonstrate that NC volume is regulated along the A/P axis during S9.

NC volume depends on the individual cytoplasmic content (the intrinsic growth) but also may depend on the transfer of cytoplasmic content from another NC through RCs (Fig 6H). The four synchronous divisions that form the 16-germline cell cyst yield eight cells with a single RC, four cells with two RCs each, two cells with three RCs each, or two cells with four RC, with one of the latter being the oocyte (Fig 6I). To determine the importance of intrinsic NC contribution in building up pressure, we measured pressure in two NCs with zero entrance

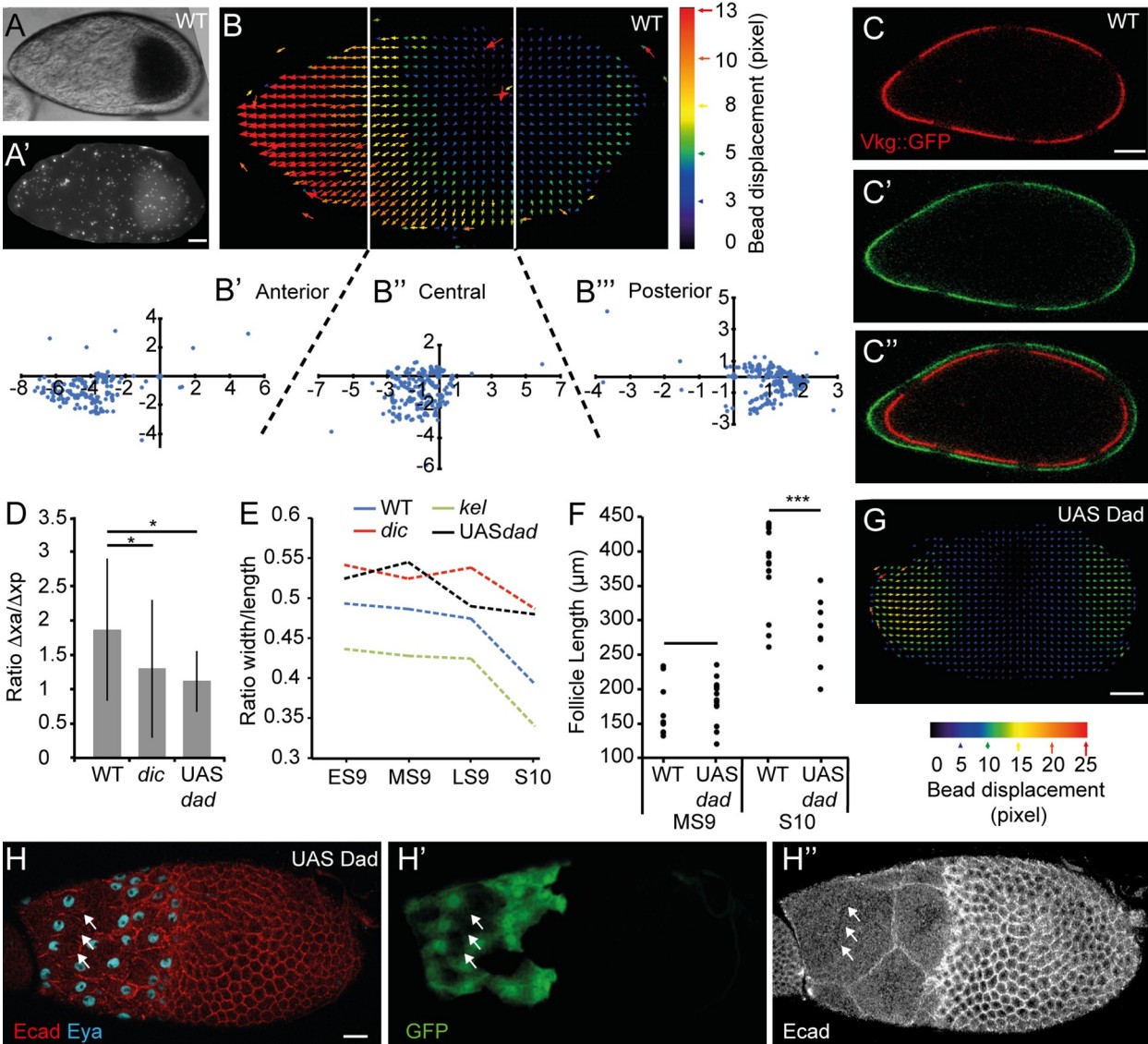

**Fig 5. The follicles grow more anteriorly than posteriorly at S9.** (A) A WT MS9 follicle covered with fluorescent beads (A'). Beads outside the follicle were manually removed to simplify the analysis. (B) Particle image velocimetry representation from the beads positioned on the follicle shown in (A) and plots presenting the coordinates of the vectors for the anterior (B'), central (B"), and posterior (B"') areas. For each area, the initial $(x, y)$ coordinates of the beads are (0,0). Each blue dot corresponds to the final $x$ and $y$ coordinates of a bead in μm. The vectors represent bead displacement (in pixel) and are color-coded as a function of their displacement. Duration of the experiment: 30 min. (C) WT MS9 follicle from a female carrying the *vkg*::*GFP* transgene to mark the BM (red in C, green in C'). Six bleached areas are visible at t = 0 (C) and at t = 80 min (C'). The two images have been overlaid by aligning the central bleached areas (C") to show follicle growth during the interval. (D) Box and whisker plot of the ratio between the anterior ("xa") and posterior ("xp") growths of the BM in WT ($n = 9$), in *dic* follicles ($n = 6$), and in Dad-expressing follicles ($n = 5$). (E) Evolution of the ratio between the width and the length of the follicles from ES9 to S10 ($n > 20$ per stage). (F) Comparison of follicle length at MS9 and at S10B between WT and Dad-expressing follicles ($n > 20$ per stage). UAS-*Dad* is expressed under the *tj*-Gal4 driver. (G) Particle image velocimetry representation from the beads positioned on a Dad-expressing follicle. Duration of the experiment: 30 min. UAS-*Dad* is expressed under the *tj*-Gal4 driver. (H) MS9 Dad-expressing follicle. Only StC express Dad (GFP) (H'). The breakage of an NC membrane (arrows) are shown in H". Scale bar: 20 μm. Data for graphs (D), (E), and (F) can be found in the S1 Data file. BM, basement membrane; *dic*, *dicephalic*; Ecad, Ecadherin; ES9, early S9; Eya, eye-absent expression; GFP, green fluorescent protein; *kel*, *kelch*; LS9, late S9; MS9, mid S9; S, stage; StC, stretched cell; WT, wild type.

RCs (ERCs) localized either in the anterior or in contact with the oocyte. The anterior NC always displays a pressure superior to the posterior NC (S8D Fig), indicating that intrinsic NC growth is an important variable in controlling the NC volume and, therefore, pressure.

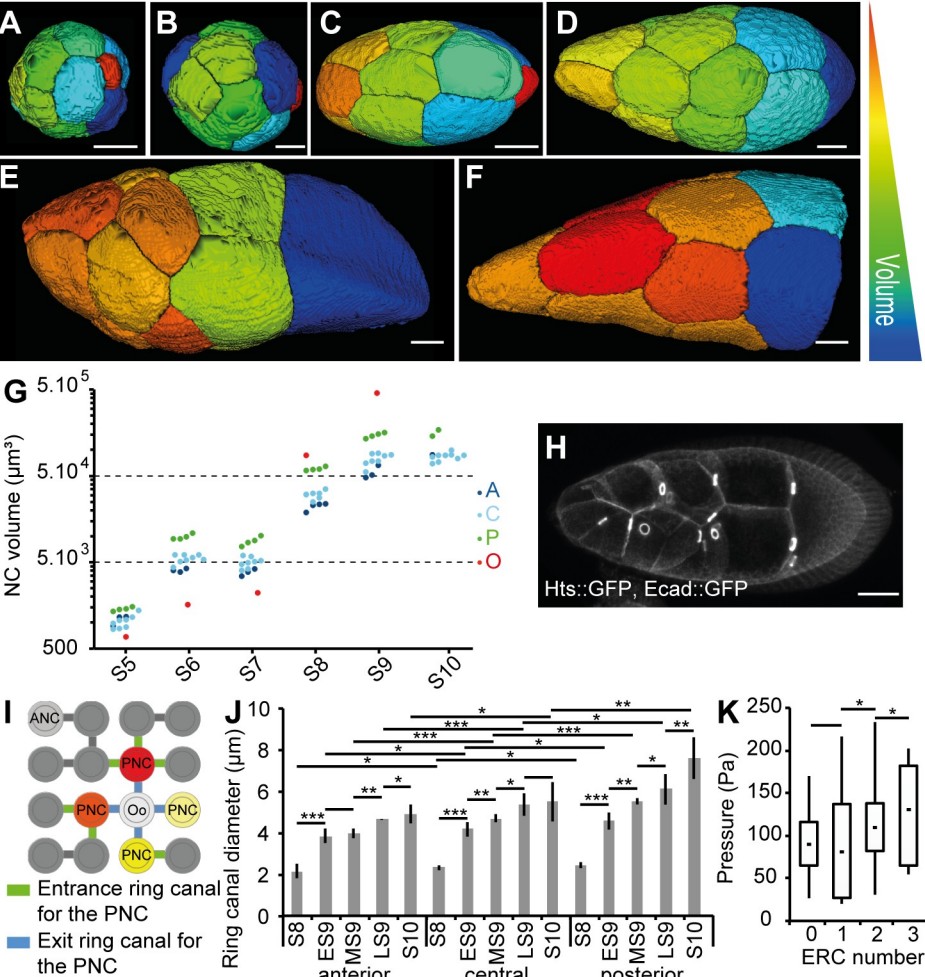

**Fig 6. Intrinsic NC growth is important for the pressure gradient.** (A-F) The 3D reconstructions of germline cells in WT follicles at S5 (A), S6 (B), S7 (C), ES9 (D), LS9 (E), and S10 (F). Only NC, but not the Oo, are shown in (F). The arbitrary color scale indicates relative cell volumes for each follicle. (G) Individual volumes of anterior ("A," dark blue), central ("C," light blue), and posterior ("P," green) NC and the Oo (red) from S5 to S10 WT follicles. For S10, the segmentation was incomplete. (H) An S9 follicle showing the RC (marked with Hts::GFP and visible as bright rings) between the NC. (I) Schematic representation of the 16 germline cells and their stereotypic connections through RC. The ANC is shown in light gray. The four PNCs (red, orange, and yellow) connect to the Oo (white) via the RC shown in blue. One PNC (light yellow) has no ERC, whereas the three others have one, two, or three ERCs (green). (J) Quantification of RC inner diameters from S8 to S10 as a function of their anterior ("A"), central ("C"), and posterior ("P") localization (*n* = 5 per stage). (K) Box and whisker plot of inner pressure in the four PNCs as a function of the number of ERCs (*n* > 10 for NCs with zero, one, or two ERCs and *n* = 5 for NCs with three ERCs). Scale bar: 20 μm. Data for graphs (G), (J), and (K) can be found in the S1 Data file. ANC, anteriormost NC; Ecad, Ecadherin; ERC, entrance RC; ES9, early S9; GFP, green fluorescent protein; LS9, late S9; MS9, mid S9; NC, nurse cell; Oo, oocyte; PNC, posteriormost NC; RC, ring canal; S, stage.

## The gradient of cytoplasmic pressure is modulated by RCs

According to Poiseuille's law describing flow of a viscous fluid through a tube, cytoplasmic flux through an RC is proportional to the difference in pressure between neighboring NC and increases rapidly with the diameter of the RC. Our analyses of NC pressure in *kel* follicles, which have previously been shown to bear smaller RC [54], demonstrate that RC size plays a role in building NC pressure. Although it is known that RC undergo an approximately 7-fold increase in diameter throughout follicle development [27], it is unclear if any differences exist

along the A/P axis. We have now carried out such measurements in the WT and confirmed that between S8 and S10, RCs do indeed undergo progressive growth. In addition, we observed a gradient of size along the A/P axis, with the smallest RC in the anterior (Fig 6J and S8C Fig). Thus, RC size may participate in establishing the pressure gradient, since narrower RCs connect anterior NCs, and wider RCs connect posterior NCs to the oocyte, which might help in building up pressure in the former and in preventing excessive buildup in the latter.

We then tested whether the number of ERCs per NC could also be related to different inner pressures. The oocyte has four ERCs, each connected to a different posterior NC. Because of the pattern of cyst formation, the four posterior NCs have either zero, one, two, or three ERCs [25]. Given what we have shown about cytoplasmic fluxes in the WT follicle, we hypothesized that the posterior NC with three ERCs would have a higher cytoplasmic pressure than the one with no ERC. In 65% of cases, the highest pressure was measured in the cell that has the highest number of ERCs (Fig 6K and S8E Fig), indicating that pressure at the same position along the A/P axis may vary as a function of the number of ERC. Importantly, the pressure in NCs with three ERCs is always lower than in more anteriorly localized cells, which in most cases bear zero ERCs. The number of ERCs is thus not responsible for establishing the gradient but likely only explains variations in gradient intensity.

## Discussion

Mechanical properties of biological elements are key components of organ morphogenesis, acting either directly by generating internal forces or indirectly by applying forces and constraints on cells. Cell mechanical properties are usually considered to be determined by cortical tension. However, another contributor is cytoplasmic pressure, which serves to counteract cortical tension. Here, we show that cytoplasmic pressure is graded within a well-defined group of NCs and that this pressure controls TGFβ expression in the surrounding epithelial cells. As TGFβ itself induces their shape changes and BM softening, we propose that in fine, both NC cytoplasmic pressure and BM softening promote enhanced anterior follicle elongation (Fig 7).

The most direct method to measure pressure is through the use of a pressure probe, but a major drawback is that it does not allow multiple measurements within a single sample. Another possibility is to deduce pressure by measuring cytoplasmic flow between NC, either

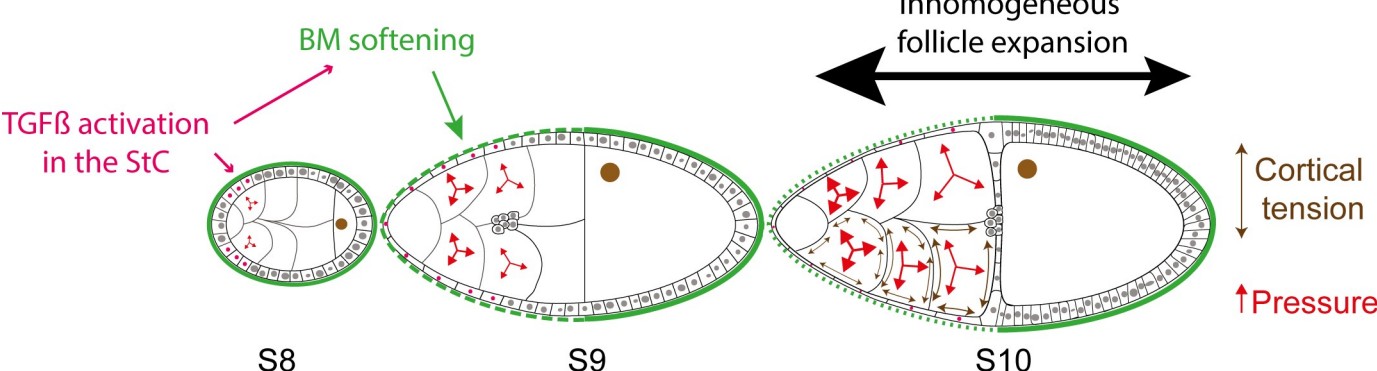

**Fig 7. Model of the role of the NC cytoplasmic pressure on StC flattening and follicle elongation.** From S8 to S10, a gradient of NC pressure (small to large red arrows) is established. When NC pressure reaches a certain level, it induces TGFβ signaling in the surrounding StC (pink), allowing their flattening. The cortical tension (double-sided brown arrow) appears to be inhomogeneous at LS9 and S10 (for simplification, it has been drawn at S10 in only half of the NC). In parallel, TGFβ controls BM remodeling and softening (dotted to plain green line) [31]. This favors anterior follicle expansion (large black arrow) versus posterior expansion (small black arrow), allowing the growth of the NC. This growth allows the maintenance of the pressure, which continues to increase during S9 and S10, under a certain threshold, preventing NC membrane breakage. BM, basement membrane; LS9, late S9; NC, nurse cell; S, stage; StC, stretched cell.

by observing the movement of cytoplasmic vesicles through RCs or by photoactivating fluorescent molecules in one cell and observing their appearance in a neighboring cell. However, we were unable to efficiently detect vesicle movement through multiple RCs along the A/P axis and phototoxicity decreased the feasibility of performing multiple measurements within a single follicle. Therefore, we chose to measure pressure using a combination of noninvasive tools that do not compromise tissue integrity while simultaneously allowing us to test whether our measurements were dependent on cortical tension.

Our first approach, AFM, makes it possible to probe deep into tissues in their native environment. Probing NC requires deforming BM, StC, NC membranes, and NC cortical actomyosin networks. Local cell integrity is preserved during experimentation, since no variation in pressure was observed over 100 measurements on a given cell. We show that our method is mostly sensitive to pressure by using different chemical treatments (NaCl treatment, collagenase) and genetic backgrounds (*dic*, *kel*, and *Pendulin*). NC pressure values are in the range observed for other animal tissues, such as chick embryo blastula (10–200 Pa; [47]) or mouse lumen blastocyst (around 300 Pa; [48]), or for cultured cells [46]. In all three studies, pressure was similarly determined via noninvasive approaches, such as AFM or micropipette aspiration. Our second approach was to generate 3D reconstructions of the germline in order to infer differential inner pressure between the NC by measuring the curvature of their membranes [34], which allowed us to confirm the AFM measurements at late S9 and S10 and to establish that the gradient is already present at S8.

Sqh activity pattern suggests the existence of a postero-anterior gradient of cortical tension only at the end of S9, which is too late to influence StC flattening. Though gradients in both pressure and curvature persist in the absence of Myosin activity, treatment with the ROCK inhibitor results in a decrease in pressure values, indicating that actomyosin activity participates in NC mechanical properties. This could be due to the contribution of either the cortical or noncortical network or both. It is not possible to evaluate the importance of cortical actomyosin activity between NC by laser ablation, since multiple ablations cannot be carried out in a single follicle. Overall, we conclude that NC mechanical properties are graded from S8 to S10 by cytoplasmic pressure with a possible late contribution of cortical tension.

Although not all the components involved in establishing the pressure gradient are known, our data clearly show the implication of intrinsic NC growth, thus supporting the concept of proliferative pressure proposed as early as 1874 [57], and of the size and pattern of the RCs. By precisely measuring NC volume, we show that individual NC growth is not regulated as a function of pressure. It is known that DNA content of the NC is inhomogeneous at S9 and S10, with more DNA in posterior NC than anterior NC [26,58]. Whether this difference in DNA content could participate in the establishment of the inner pressure gradient remains to be tested. Difference in the spatial distribution of amiloride-sensitive Na+,H+ exchangers and Na + channels are also likely to play a role in controlling NC volume. In 2015, Kruger and Bohrmann described that the amiloride-sensitive Na+ transporters are distributed in an A/P gradient within the NC group, with more transporters in anterior NC compared to posterior [59]. Although drastic modifications of such channel activities might be deleterious for follicle integrity, analyzing NC volume differences in conditions mildly affecting their activity could be informative.

Our observation of a pressure gradient raises the question of how it can be maintained without certain NCs growing at the expense of others or emptying their contents through RC. A similar question was recently investigated theoretically and experimentally in plant tissues [60,61]: plasmodesmata and the cell wall play the same role as RC and the actomyosin cortex, respectively. Extrapolating from those studies, it would be expected that NCs maintain their relative volumes whenever actomyosin cortex viscosity and RC hydraulic conductivity are low

enough, so that the movement of water and cytoplasm is partially limiting changes in cell volumes.

Our data also suggest that RCs play dual and opposing functions in different NCs: the large exit RC between the posterior NC and the oocyte may help prevent high pressure from building in the posterior, whereas the small RC in the anterior NC may help increase pressure there. The number of ERCs also plays a role in building pressure in certain posteriorly localized NCs, as NCs with such RCs are mainly in the central and posterior areas [62].

StC flattening depends on both TGFβ activity and germline growth [32,63]. TGFβ activity is required in the StC to set up the actomyosin pattern that leads to AJ remodeling. *Mad* or *tkv* somatic clones lead to the presence of small and insufficiently flattened StC [63], indicating that NC pressure is not sufficient to induce cell shape changes in the absence of TGFβ activity in the StC. By analyzing *dic* and *kel* follicles, we now demonstrate that NC inner pressure controls the triggering as well as the progression of StC flattening by acting on TGFβ signaling. Kolahi and colleagues had previously shown that the degree of StC size depends on NC growth [32]. Nevertheless, a strict correlation between NC pressure and pMad levels during follicle growth cannot be drawn, possibly because once the TGFβ pathway is activated by NC pressure, a negative feedback on the pathway—for instance, through Dad activity—may kick in and lead to reduced expression of pMad, despite the continued presence of high NC pressure. It has been shown that Yorkie (Yki), a transcriptional coactivator, is present in the StC [64]. Yki is known to stimulate cell growth with DNA-binding partner proteins, including Mad [65], and to be regulated by cortical tension [66]. Future experiments will determine whether or not the molecular links between NC pressure and StC flattening requires TGFβ and Yki.

Several studies have shown the importance of BM structure and stiffness in the elongation of the follicle and the egg [49,67,76–79,68–75]. Two nonexclusive models have been proposed: first, that a softer BM at both poles of the follicle would favor expansion along the A/P axis [51] and, second, that the structure of the BM, with stiff fibril-like structures oriented perpendicular to the A/P axis, would prevent radial expansion [31]. The discovery of a pressure gradient within the NC reveals a third mechanism that acts from S8 to S10 (Fig 7). At S8, pressure increases in the anterior NC and induces TGFβ in the StC, which in turn leads to flattening and to the local softening of the BM. On the one hand, StC flattening likely facilitates NC growth, which is required to build up the pressure, by allowing a rapid transfer of oxygen and nutrients from the hemolymph. On the other hand, the softening of the BM above the StC likely helps the increase of NC volume and the maintenance of inner pressure under a certain threshold. Together, these interconnected genetic and mechanical regulations may allow the coordination in growth and shape of two adjacent populations (StC and NC) to shape yet another entity, the oocyte.

Our work highlights the role of fluid pressure in development, which has begun to be increasingly recognized [80]. Our results show that the ovarian follicle can be considered as a pressurized shell, with the BM under pressure from NC and oocyte. This echoes experimental findings that adult *C. elegans* [81] and regenerating hydra [82] may be modeled as shells under pressure. We find that follicle growth is driven by cell pressure, similarly to regenerating hydra [82], and to elongation and inflation of the notochord in the *Xenopus* embryo [83,84]. It is worth noting, however, that cell-cell pressure differentials have been entirely overlooked so far. Our data show that an A/P gradient in cytoplasmic pressure is important for follicle morphogenesis. One area that has been more explored is that of pressure caused by fluid accumulating within a lumen, such as in tubulogenesis during lung and kidney development [80]. Two recent studies reported the key role of lumen pressure in the mouse blastocyst in determining embryo size and cell fate, as well as in establishing the first axis of symmetry [48,85]. More broadly, the implication of osmotic pressure in growth or morphogenesis is also well

documented in walled cells such as in plants, fungi, or bacteria [45,86]. Importantly, some of these models also point out the importance of the balance between fluid pressure and extracellular constraints in sculpting organs, a balance that we also evidence in an animal system.

In summary, our work highlights the importance of the mechanical properties of cells or tissues neighboring a tissue undergoing a morphogenetic event and reveals the mechanical role of cytoplasmic pressure in shaping cells and organs.

## Materials and methods

### Fly stocks and clones generations

Canton S was used as WT; the other fly stocks are: *hts*::*GFP* [87], *Ecad*::*GFP* [88], *PHPLCγ*::*GFP* [89], *sqh*::*mCherry* [9], *P(UAS-tkv$^{Q199D}$)* (referred to as TkvA), *P(UAS-Dad.T)* [90], *traffic jam-Gal4* flies [91], *dic$^1$* [53], *kelch$^{ED1}$* [54], *Pen$^{D14}$* [56], *sqh$^{HMS00437}$*, *sqh$^{HMS00830}$* [92], w*; P{matα4-GAL-VP16}V37 (that we refer to as matα4-G4) [93], P{nos-GAL4.U} that we refer as nos-G4 [94], P{w[+mC] = otu-GAL4::VP16.R}1, w[*]; P{w[+mC] = GAL4-nos.NGT}40;P{w[+mC] = GAL4::VP16-nos.UTR}CG6325[MVD1] that we refer as MTD-G4 [95], and *vkg$^{G454}$* (allele containing a GFP protein trap in the Col IV α2 chain Viking that we refer as Coll IV::GFP in the text [96]).

Fly stocks were cultured at 25°C on standard food. Ectopic expression of TkvA or Dad were performed by generating Flip-out Gal4 clones in animals carrying the hs-FLP22 and the AyGAL4 UAS-GFP transgenes [97] or by crossing with *tj*-Gal4 flies. Flippase expression was induced by heat-shocking 3-d-old females at 37.3°C for 1 h to generate Flip-out clones. Adult females were fed on abundant yeast diet for 2–3 d prior to dissection.

### Follicle staining and staging

Ovaries from females were dissected directly into fixative 3–4 d after Flippase induction and stained following the protocol described in [28]. To avoid fluctuations of the depth of the follicles that are squeezed by the coverslip, each slide contains 15 ovaries, from which S11–S14 are removed. After dissection of the follicles, most of the PBS is removed and 20 μl of the imaging medium (PBS/Glycerol [25/75] [v/v]) is added before being covered by a 22/32-mm coverslip. The following antibodies were used: goat anti-GFP (1:1,000; Abcam), rat anti-Ecad (1:200; Developmental Studies Hybridoma Bank), rabbit anti-pMad (1:200; Nakao and colleagues, 1997), mouse anti-Phospho-Sqh (1:1,000; Cell Signaling).

WT follicles were staged as a function of their full length (Lf), the oocyte length (Lo), and the timing of the different morphogenetic processes, including border cell migration and StC flattening, that occur at S9 [25,26]. Usually, early S9 (ES9) follicles have an Lf of 205 ± 15 μm with row1 StC of about 100 μm$^2$ of apical area; MS9 follicles of 240 ± 20 μm with row1 StC of about 200 μm$^2$; and LS9 follicles of 270 ± 20 μm with row1 StC of about 350 μm$^2$. To stage *dic* and *kel* follicles, we first sorted them into ES9, MS9, or LS9 based on the progression of StC flattening. For *dic*, the start of ES9, MS9, and LS9 were ascribed based on the presence of StC with mild (about 50 μm$^2$), medium (about 150 μm$^2$), and severe (>250 μm$^2$) flattening, respectively. For *kel*, the start of ES9, MS9, and LS9 where ascribed when row 1 StC showed mild (about 100 μm$^2$), medium (about 200 μm$^2$), and severe (above 350 μm$^2$) flattening, respectively. We next plotted follicle length for each of the three groups to define the relationship between these two variables (S5 Fig) and used both to stage the follicles. Accordingly, follicles with highly abnormal values of length in regards to StC flattening were excluded from our studies.

## Fluorescent quantifications

Mutant and *viking*::*GFP*/CyO (which served as a control) flies were mixed before dissection. Both WT and mutant ovaries are thus fixed, stained, and mounted together. Mutant and control follicles were discriminated thanks to the specific GFP accumulation in the BM in *vkg*::*GFP* follicles.

For pMad quantification, a projection of all of the z sections in which StC nuclei are visible is made and background was subtracted. Nuclei were individually selected and registered as regions of interest (ROIs). For each ROI, the "area" and the "integrated density" were quantified. Measurements were also performed in three areas located close to nuclei to estimate background and a mean of fluorescent background is calculated. The corrected total nuclei fluorescence (CTNF) for each nuclei is given by using the formula: CTNF = integrated density − (area of selected nuclei × mean of background fluorescence).

For Ecad, Sqh, and pSqh quantifications, measurements were performed on the focal plane where the border cells are visible. ROIs of 2 μm² were positioned on plasma membrane based on the Ecad staining. The ROI measurement tool was then used to calculate the mean gray values for each area and the ratio was calculated.

## AFM

AFM indentation experiments were carried out with a Catalyst Bioscope (Bruker Nano Surface, Santa Barbara, CA) that was mounted on an optical macroscope (MacroFluo, Leica) using an objective (10× objective, Mitutuyo). A Nanoscope V controller and Nanoscope software versions 8.15 and 9.2 were utilized. All quantitative measurements were performed using nitride cantilevers with silicon pyramidal tips (SCANASYST-FLUID, Bruker AFM probes) with a nominal spring constant of 0.7 N/m and a nominal tip radius of 40 nm. The actual spring constant of cantilevers was measured using the thermal tuning method [98,99] and ranged from 0.6 to 0.9 N/m, which was sufficient to indent the sample without damaging it. The deflection sensitivity of cantilevers was calibrated against a clean silicon wafer. Fresh dissected follicles were fixed on a petri dish coated with poly-L-lysine (0.5 mg/ml) and were covered by living medium. Follicles are kept for an hour maximum before being discarded. All experiments were made at room temperature and the standard cantilever holder for operation in liquid was used. The petri dish was positioned on an XY motorized stage and held by a magnetic clamp. Then, the AFM head was mounted on the stage and an approximated positioning with respect to the cantilever was done using the optical macroscope.

To record force curves, the Ramp module of the Contact mode in fluid was used. With this module, individual force curves are acquired at discrete points chosen using the optical image of the follicle. Each AFM measurement consists of the acquisition of 100 force curves extracted as 10 × 10 matrices with indentation points spaced 100 nm apart.

The pressure was deduced from local follicle stiffness and geometry [45,100]. The local stiffness, *k*, is derived from the force–indentation curves by fitting to a linear model, using depths between 1 and 2 μm in order to probe the pressure of the NC while minimizing the influence of the BM and of the StCs; the stiffness was obtained in piconewtons per meter (pN/m). The geometry at the indented location was characterized by the radii of curvature $R_1$ and $R_2$. Considering the follicle to be approximately a surface of revolution, $R_1$ is the local radius of curvature of the follicle outline and $R_2$ is the distance along the normal to the outline of the indented point to the axis of revolution. Radii were measured (in μm) from the optical images (from the macroscope) using ImageJ.

The pressure was then computed using the following equation [100]:

$$P = k/\pi \, 2R_1R_2/(R_1 + R_2),$$

with values in Pascals (Pa). The measurements referred to as "anterior" were not necessarily performed on the most anterior cell of the follicle, as the curvature of the follicle can sometimes prevent access to this cell. This difficulty occurs when probing certain LS9 (depending on the angle they form with the petri dish, with the anterior extremity tilted up or down), and most of the S10 follicles. In such cases, the probe is usually able to reach an area located about 30 μm from the most anterior point.

To reduce variability, each experimenter performed WT and mutant measurements and comparisons were made between sets obtained by the same experimenter.

## Bespoke protocol to probe internal NC pressure

Living follicles were dissected as previously described [31] and cultured in poly-L-lysine–coated petri dishes (S1D Fig). Measuring the apparent stiffness of NC requires deforming both the BM and the follicular cells. We and others have previously established that the BM is on average 0.2 μm thick, whereas follicular cell height varies from about 5 μm before S9 to less than 1 μm after flattening [31,32]. Since AFM measurements typically allow a maximum of 3–4 μm indentation, we could not probe NC prior to S9 (S1E Fig). During S9, we probed one, two, or three areas, located above the anterior, central, or posterior regions of the NC, respectively, depending on the position of the wave of flattening (S1F Fig). No measurements were performed when StC height was superior to 1 μm. To be mostly sensitive to NC pressure, the stiffness of the cantilever and the force applied were chosen to enable indentation depths of 1.5–2.5 μm, which is more than the thickness of the BM and of the flattened StC combined. To perform the measurements, we used the Contact mode of the AFM and registered 100 raw force curves from a $10 \times 10$ matrix, with indentation points spaced 100 nm apart.

## Justification of pressure deduction from AFM

In order to interpret AFM stiffness, we first sought the mechanical model that would best represent follicles. NCs bulge out following collagenase treatment, showing that the BM significantly contributes to constraining NC shape. Accordingly, we used a comprehensively investigated model [100] and considered the follicle as a thin shell, corresponding to the BM, under (possibly nonuniform) pressure due to NCs. We used indentation depths (1.5–2.5 μm) much greater than BM thickness (0.2 μm); in this regime, Ref. [92] established that stiffness is only related to pressure and to local curvatures of the surface, as given in the equation above, accounting for the local ellipsoidal shape of the follicle. We next wondered how local the deduction of pressure was and which cells we were probing. To address this, we first used depths greater than StC height. Second, the scale of measurement is given by $(d\,R_\mathrm{m})^{1/2}$ [100], where $d \sim 2$ μm is the typical indentation depth and an $R_\mathrm{m}$ of ~50 μm is a typical value of follicle curvature, yielding a scale of about 10 μm, which is larger than StC height and smaller than the size of the NC. Therefore, the measurement is sufficiently local to assess single NCs. To assess pressure measurements, we treated follicles with solutions with higher concentrations of osmolytes, which reduced AFM-based pressure as expected. Altogether, we find differences in pressure ($\Delta P$) between NCs of about 50 Pa, with higher pressures in anterior NC, consistent with their convex posterior membranes. Based on the typical cell-cell radius of curvature observed ($R \sim 50$ μm), we obtain values of cortical tension $2R\Delta P \sim 5$ mN/m, which is comparable to values of the order of 10 mN/m found in single animal cells (see, e.g., [101]). We thus conclude that our measurements mostly reflect the pressure of individual NCs.

## Chemical treatment of the follicles

Collagenase (1,000 units/ml CLSP; Worthington Biochemical Corp) was added to a final volume of 200 μl. The reactions were stopped with 10 mM L-Cysteine after 80 min of incubation.

These experimental conditions lead to collagen digestion all around the follicles (see Chlasta and colleagues, 2017 [31], for details).

The ROCK inhibitor Y-27632 (Sigma-Aldrich) was added to a final concentration of 100 μM.

The Latrunculin B drug (Sigma-Aldrich) was added to a final concentration of 125 μM and measurements were performed after a treatment of 20 min [50].

### Imaging for 3D reconstruction

The MARS pipeline is based on the fusion of several confocal stacks of images, taken with multiple angles, in order to recreate a highly resolutive 3D reconstitution of the object. Segmentation is performed from the fused stack (for details, see [52]).

PHPLCγ::GFP follicles were dissected in PBS1X, before being stuck onto an coverslip coated with poly-L-lysine (0.5 mg/ml), which was glued on a Pasteur pipette. The pipette was fixed in a large petri dish, allowing its rotation in three angles (−30˚, 0˚, and +30˚). Z-stacks were taken from the three angles using a 40×, 0.75 NA water-immersion objective of a Zeiss LSM700 confocal microscope (for S4 to M9 follicles) or using the 32×, 0.85 NA water-immersion objective and a pulse infrared laser (Chameleon OPO) of a Zeiss LSM 710 (for thick follicles starting mid-late S9). The follicles were either imaged in living conditions or fixed in a 4% formaldehyde solution and anti-GFP antibodies were used to detect PHPLCγ::GFP expression.

### Cell curvature measurements

The 2D measurements are performed on fixed ovaries by manually applying circles fitting the curvatures between adjacent NC at a focal plane allowing border cell visualization.

The 3D measurements are performed from segmented NC generated by the MARS method using a custom-made Image macro, called "Find-Curve." The Find-Curve macro for the ImageJ program [102] automatically processes all cell stacks contained in a root folder indicated by the user. As the third dimension of the stacks, created by the MARS 3D reconstructions, corresponds to the $z$ axis, the program generates (X and Y) complementary orthogonal views. The three different views are then independently analyzed. Using the "Default" threshold, derived from the Iso-Data algorithm [103], the macro identifies the slice with the largest area. An option has been implemented to manually specify the slice to analyze. On this slice, for every point (O) of the perimeter, the angle (OA, OB) is calculated, with A and B two neighbor (10th degree) points, respectively upstream and downstream to O. The determination of local maximal angle values along the perimeter allows identification of the "summits" of the cell. These summits are used to define the various segments of the cell and generate the fitting regular polygon. For all segments, the radius of the best-fitting osculating circle is calculated [104,105]. This value is the curvature radius ρ of the studied segment. Finally, an.html report file is automatically created for human quality control.

A folder, called "test stacks for Fig 2G," containing the three stacks corresponding to the NCs presented in Fig 2G and the macro can be downloaded by following this link: https://gitbio.ens-lyon.fr/dcluet/Find_Curve.

### Flow velocity measurements

Flies were dissected in PBS 1X and placed in a petri dish coated with poly-L-lysine and incubated 10 min with fluorescent beads (Fluoresbrite Multifluorescent Microspheres 1.00 μm 2.5% [Polysciences], dilution 1/100) before being rinsed and maintained in the living medium described in Prasad and colleagues 2007 [106]. Images were acquired with Leica Macrofluo Plan Apo 5.0×/0.50 LWD objective. Image stacks (z = 80 μm) were taken every 30 min and

analyzed with an ImageJ plugin, https://sites.google.com/site/qingzongtseng/piv [107]. Beads outside the follicle were manually removed to simplify the analysis. The anterior, central, and posterior areas are defined by dividing the length of the follicle.

### BM photobleaching

Homozygous flies for *vkg*::*GFP* were dissected and maintained in the living medium [106]. Follicles were placed in a coverslip coated with poly-L-lysine, which was glued to the bottom of a custom-made open chamber. Living medium was added from the top and covered by semipermeable membrane. Fluorescent images of the samples were acquired on an inverted Zeiss LSM 710 confocal microscope with 40×/0.75 water-immersion objective using the 488-nm line of an Argon Laser at 25˚C. All images were acquired at a 512 × 512 pixels resolution. FRAP experiments were carried out by scanning over 20 μm the follicle starting at about 15 μm from the coverslip. z Sections were performed with a step of 1 μm. The size of bleached ROI was about 10 × 10 × 10 μm, centered within the 20-μm-sized scanned area. Significant bleaching occurs after 20 iterations. All images were acquired at a scan speed of 4. Stacks of images were taken each 30 min over 2 h.

### Statistical analyses

Normality of the samples was tested using Agostino-Pearson or Shapiro's tests. Student *t* test, Mann-Whitney test, or Wilcoxon test were performed, depending on the normality and the variances.

### Supporting information

**S1 Data. Excel spreadsheet containing, in separate sheets, the numerical data for figure panels 1E, 1G, 1I, 2A, 2B, 2C, 2D, 2F, 2H, 3D, 3E, 3F, 3K, 3L, 4M, 4N, 5D, 5E, 5F, 6G, 6J, 6K, S2A, S2B, S2D, S2F, S2G, S2H, S3B, S3C, S4C, S4E, S4F, S4G, S4H, S4I, S6A, S6E, S6G, S6H, S7J, S7K, S8A, S8B, S8C, S8D, and S8E.**
(XLSX)

**S1 Movie. Fused stack and 3D reconstruction of the germline of the S9 follicle presented in S4A Fig.**
(MOV)

**S2 Movie. The 3D reconstruction of the germline and the border cells of the S9 follicle presented in S4A Fig.** The second part of the movie shows the border cells and the imprints of the posterior NC in the oocyte. The imprints are concave, indicating that the posterior nurse cells bulge toward the oocyte.
(MP4)

**S1 Fig. Method for AFM measurements.** (A-C) Sections through WT live follicles (A-B) or fixed follicles (C). (D) Diagram of the setup used to perform AFM measurements. Living follicles are stuck to a poly-L-lysine–coated petri dish filled with culture medium. The AFM comprises a cantilever with a tip that is used to probe the follicles. The bending of the lever when it encounters the follicle is detected by a photodiode that captures changes in a laser beam reflected onto the upper face of the cantilever. (E) Schematic representation of the probed area: the tip deforms the basement membrane (green) and the StC (red). In StC that have already flattened (E'), the underlying nurse cells (brown) are also deformed, allowing measurements. (F) Schematic representation of a LS9 follicle and of the cantilever. At this stage, only two regions were probed: above the anterior ("A") and central ("C") nurse cells. The posterior

nurse cells are probed only in S10 follicles. For each region, a $10 \times 10$ matrix is measured, with indentation points spaced 100 nm apart. (G) Force–indentation depth curve obtained on a follicle with a pyramidal probe tip. The curve is fitted using the linear model (red line) to obtain the elastic modulus. Only the zone of interest (−1 to −2 μm) is fitted. (H) Schematic representation of the geometric measurements taken at areas where inner pressure was measured with AFM. Two circles are drawn to fit either the entire follicle (dotted black line) or the AFM-probed area (dotted blue line at red dot), and the two radii, R1 and R2, are used to calculate inner pressure (see Methods). Scale bar: 20 μm. AFM, atomic force microscope; LS9, late S9; S, stage; StC, stretched cell; WT, wild type.
(TIF)

**S2 Fig. Control experiments for the AFM measurements.** (A) Inner pressures of anterior ("A"), central ("C"), and posterior ("P") NCs in individual WT follicles at S9 and S10. (B) Box and whisker plots of the 100 measurements taken in anterior ("A"), central ("C"), and posterior ("P") NCs from a S10 follicle. (C) Color-coded representation of the 100 measurements in a single probed region of a WT S10 follicle. The color-coded scale is shown. (D) AFM measurements of inner pressure in WT LS9 and S10A follicle before and after Latrunculin B treatment ($n$ = 10). (E) WT S10 follicle before (E) and after (E') NaCl treatment. (F) Force–indentation depth curves from anterior, central, and posterior NCs of a WT S10 follicle before (solid lines) and after (dotted lines) NaCl treatment. (G) Fold-change of inner pressure following osmotic treatment shown as a function of solution osmolarity (NaCl 0.5 M in water–NaCl 0.5 M in PBS–Sorbitol 1 M in PBS–NaCl 1 M in PBS from the lowest to the highest osmolarity). No significant influence of the position of the NC (anterior, central, or posterior) has been observed. (H) AFM measurements of inner pressure in anterior, central, or posterior NCs after collagenase treatment. Scale bar: 50 μm. Data for graphs (A), (B), (D), (F), (G), and (H) can be found in the S1 Data file. AFM, atomic force microscope; LS9, late S9; NC, nurse cell; S, stage; WT, wild type.
(TIF)

**S3 Fig. Methods for NC curvature measurements in 2D and for NC 3D reconstruction.** (A) Schematic representation of a WT S9 follicle representing the method used to measure NC curvature in fixed follicles: circles (blue dotted line) are apposed to fit a particular NC membrane and the radius (r) is measured. (B) Box and whisker plots of radii of the membrane curvature for anterior ("A"), central ("C," both the anteriormost "Ca" and the posteriormost "Cp"), and posterior ("P") NCs in WT ES 9 to S10 follicles ($n$ = 98 cells for ES9; 42 for MS9; 25 for LS9; and 15 for S10). (C) Percentage of convex (orange) and concave (blue) NC posterior membrane curvatures at different stages in presence or not of the ROCK inhibitor (Y-27632) for WT follicles. (D) WT follicle after addition of ROCK inhibitor. (E) Schematic representation of sample acquisitions for the MARS method. (F) The 2D surface projections of a WT S8 follicle imaged at three different angles. Reference points (red dots) are used to fuse the stacks. (G) A mid-Z-slice through a 3D reconstructed follicle where the three individual image stacks were fused into a single high-resolution stack. (H) The 3D segmentation of the reconstructed follicle (G) showing only the germline cells. Scale bar: 20 μm. Data for graphs (B) and (C) can be found in the S1 Data file. ES9, early S9; LS9, late S9; MS9, mid S9; NC, nurse cell; S, stage; WT, wild type.
(TIF)

**S4 Fig. NC membrane curvature measurements from 3D reconstruction.** (A) A mid-Z-slice through a 3D reconstructed mid S9 follicle. (A'-A"') Segmented germline cells visualized from the $z$ (A'), $y$ (A"), or $x$ (A"') axis. (B) The S8 follicle presented in Fig 2G viewed at a different z

section, along with the corresponding 3D segmentation (B'). $B_A$, $B_{Ca}$, $B_{Cp}$, and $B_P$ show to slices with the largest area in the 3D segmentation of the NC under study, which are labeled as in Fig 2G. The (B') panels present the outlines (solid white lines) of the same cells, as well as the circles (dotted white lines) fitting the posterior membranes (blue lines). (C) Percentage of anterior ("A"), posterior ("P"), and lateral convex NC membranes in WT S5 to S10 follicles. (D) Slice through a 3D reconstructed S6 follicle along with the 3D segmented image of its germline cells (D'). The slice with the largest area within the segmented oocyte in the $z$ axis is shown ($D_{Oo}$) alongside its outline (solid white line) and the circles fitting the anterior membranes (blue line) ($D'_{Oo}$ to $D''_{Oo}$). (E) Percentage of convex (orange), flat (gray), and concave (blue) NC posterior membrane curvatures in function of the position of the NCs along the A/P axis (anterior ["A"], central ["C"], and posterior ["P"]) in WT S5 to S10 follicles ($n$ comprised between 9 and 49 cells). (F-H) Percentage (F, G) or number (H) of convex (orange), flat (gray), or concave (blue) curvatures of anterior NC membranes (F, $n > 60$), lateral NC membranes (G, $n > 100$), or anterior oocyte membranes (H) in WT S5 to S10 follicles. Two or three follicles were reconstructed and segmented per stage. (I) Percentage of convex (orange), flat (gray), and concave (blue) NC posterior membrane curvatures at different follicular stages in WT or in follicles with reduced *sqh* germline activity ($n > 167$). Scale bar: 20 μm. Data for graphs (C), (E), (F), (G), (H), and (I) can be found in the S1 Data file. NC, nurse cell; S, stage; *sqh*, *spaghetti squash*; WT, wild type.
(TIF)

**S5 Fig. Staging of WT, *dic*, and *kel* follicles.** (A–I) Follicles of ES9, MS9, or LS9 in WT, *dic*, or *kel* follicles. For each stage and mutant, a section through the middle of the follicle and a projection of all the sections where StCs are visible are presented. Follicle lengths are indicated. (J) Correspondence between developmental stages (ES9, black; MS9, dark gray; LS9, light gray) and follicle length. *dic*, *dicephalic*; ES9, early S9; *kel*, *kelch*; LS9, late S9; MS9, mid S9; StC, stretched cell; WT, wild type.
(TIF)

**S6 Fig. NC membrane curvature and StC flattening in *dic* and *kel* follicles.** (A) Curvature orientation in WT, *dic*, or *kel* follicle during S9 and S10 in presence or not of the ROCK inhibitor (Y-27632) ($n > 10$ follicles per stage). (B–D) Sections through *dic* live follicles (B–C) or *kel* live follicles (D). (E) Apical StC area along the antero-posterior axis, from row 1 (anterior) to row $n$ (posterior), of WT (blue), *dic* (red), and *kel* (green) MS9 follicles ($n > 10$ cells for each row). (F) Projection of all the sections where StC are visible in a S10 *dic* follicle. Most of the AJs are still visible (F'). (G, H) Apical StC area of row 1 (G) or row 2 (H), in function of WT (blue), *dic* (red), and *kel* (green) follicle length ($n > 10$ cells for each row). Scale bar: 20 μm. Data for graphs (A), (E), (G), and (H) can be found in the S1 Data file. AJ, adherens junction; *dic*, *dicephalic*; *kel*, *kelch*; MS9, mid S9; S, stage; StC, stretched cell; WT, wild type.
(TIF)

**S7 Fig. PIV representation for WT and mutants follicles.** (A) LS9 follicle with fluorescent beads at two different time points (A, A1). (A') and (A1') are enlarged views of the yellow boxes drawn in (A) and (A1), respectively. (B) The S9 follicle presented in (A). (C) PIV representation from the beads positioned on the follicle presented in (A). (D) PIV representation from the beads positioned on an ES9 follicle and plots presenting the coordinates of the vectors for the anterior (D'), central (D"), and posterior (D"') regions. For each area, the initial ($x$, $y$) coordinates of the beads are (0,0). Each blue dot corresponds to the final $x$ and $y$ coordinates (in μm) of a bead. I PIV representation from the beads positioned on a S9 *dic* follicle. (F, G, H, I) LS9 follicles from WT (F), *dic* (G), *kel* (H), or Dad-expressing (I) females with the length

and the width of the follicles indicated. In (I), UAS-Dad is expressed under the *tj*-Gal4 driver. In (H), the follicular epithelial above the oocyte is collapsing in the oocyte, possibly because of the lack of pressure in the latter. (J) Egg shape from WT females or females expressing Dad in the follicular cells under the *actin* promoter (large Flip-out clones) ($n > 40$ per genotype). (K) Pressure in anterior nurse cells in WT and in Dad-expressing follicles at MS9. UAS-Dad is expressed under the *tj*-Gal4 driver. Scale bars: 20 μm. Data for graphs (J) and (K) can be found in the S1 Data file. *dic*, *dicephalic*; ES9, early S9; *kel*, *kelch*; LS9, late S9; MS9, mid S9; PIV, particle image velocimetry; S, stage; WT, wild type.
(TIF)

**S8 Fig. Global volume of the NC and difference of pressure between NC in function of the number of RCs.** (A) Volumes of anterior ("A," dark blue), central ("C," cyan), and posterior ("P," light blue) NC and the oocyte (red) from S5 to S10 WT follicles. Central NC are further subdivided into those abutting the Ca and those abutting the four Cp. (B) Volume of the oocyte from WT S5 to S9 follicles ($n > 2$ per stage). (C) Ratios of RC diameters at different follicular stages. Ratios are presented for A/C NC, A/P NC, or C/P NC. (D) Ratio of pressure between the ANC and the oocyte-connected PNC 1RC NCs ($n = 4$). (E) Inner pressure in two posterior NCs in WT S10 follicles. The number of ERCs of the two probed NCs is indicated by a symbol (diamond, triangle, circle, or square). Data for graphs (A), (B), (C), (D), and (E) can be found in the S1 Data file. 1RC, single RC; A/C, anterior versus central; A/P, anterior versus posterior; ANC, anteriormost NC; C/P, central versus posterior; Ca, anterior NC; Cp, posterior NC; ERC, entrance RC; NC, nurse cell; PNC posteriormost NC; RC, ring canal; S, stage; WT, wild type.
(TIF)

## Acknowledgments

We thank DSHB and Bloomington Stock Center for flies and reagents. We thank Lyon Multi-scale Imaging Center (LyMiC) and Arthrotools of the SFR Bioscience (UMS3444/US8). We are very grateful to A. Guichet, D. Bilder, Anne Ephrussi, J. R. Huynh, B. Loppin, and Sally Horne-Badovinac for flies; V. Van de Bor for flies and discussions; and F. Schweisguth for discussions. We acknowledge X. Wang for critical comments on the manuscript.

## Author Contributions

**Conceptualization:** Arezki Boudaoud, Muriel Grammont.

**Data curation:** Muriel Grammont.

**Formal analysis:** Pascale Milani, David Cluet, Arezki Boudaoud, Muriel Grammont.

**Funding acquisition:** Arezki Boudaoud, Muriel Grammont.

**Investigation:** Laurie-Anne Lamiré, Pascale Milani, Gaël Runel, Leticia Arias, Stève de Bossoreille.

**Methodology:** Pascale Milani, Gaël Runel, Pradeep Das, Arezki Boudaoud.

**Project administration:** Arezki Boudaoud, Muriel Grammont.

**Software:** Gaël Runel, Annamaria Kiss, Blandine Vergier, Pradeep Das, David Cluet.

**Supervision:** Arezki Boudaoud, Muriel Grammont.

**Validation:** Pascale Milani, Pradeep Das, David Cluet, Arezki Boudaoud, Muriel Grammont.

**Visualization:** Muriel Grammont.

**Writing – original draft:** Laurie-Anne Lamiré, Muriel Grammont.

**Writing – review & editing:** Stève de Bossoreille, Arezki Boudaoud, Muriel Grammont.

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
