## [Editor Report · Decision Letter 0]

2 Jan 2020

Dear Dr grammont, 

Thank you for submitting your manuscript entitled "Gradient in cytoplasmic pressure in the germline cells controlsoverlying epithelial cell morphogenesis" for consideration as a Research Article by PLOS Biology. Thank you also for your patience while we completed our editorial process.

Your manuscript has now been evaluated by the PLOS Biology editorial staff as well as by an academic editor with relevant expertise and I am writing to let you know that we would like to send your submission out for external peer review.

Please re-submit your manuscript within two working days, i.e. by Jan 06 2020 11:59PM.

Kind regards,

Ines

--

Ines Alvarez-Garcia, PhD

Senior Editor

PLOS Biology

Carlyle House, Carlyle Road

Cambridge, CB4 3DN

+44 1223–442810

---

## [Decision Letter · Decision Letter 1]

19 Feb 2020

Dear Dr Grammont,

Thank you very much for submitting your manuscript entitled "Gradient in cytoplasmic pressure in the germline cells controls overlying epithelial cell morphogenesis" for consideration as a Research Article at PLOS Biology. Thank you also for your patience as we completed our editorial process, and please accept again my sincere apologies for the delay in providing you with our decision. Your manuscript has been evaluated by the PLOS Biology editors, an Academic Editor with relevant expertise, and by three independent reviewers.

As you will see, the reviewers find the conclusions of your manuscript interesting and significant for the field, however they also each raise several issues that need to be addressed before we can consider further the manuscript. The reviewers have concerns with the AFM data, feel that some of the data are overinterpreted and highlight limitations with the experimental approach. While it would be nice if you address Rev. 1's Point 8, we feel that it would probably require a lot of work, thus we won't insist on it to consider the manuscript for publication.

In light of the reviews (attached below), we will not be able to accept the current version of the manuscript, but we would welcome re-submission of a much-revised version that takes into account the reviewers' comments. We cannot make any decision about publication until we have seen the revised manuscript and your response to the reviewers' comments. Your revised manuscript is also likely to be sent for further evaluation by the reviewers.

We expect to receive your revised manuscript within 2 months. 

**IMPORTANT - SUBMITTING YOUR REVISION**

*Re-submission Checklist*

*Published Peer Review*

*PLOS Data Policy*

*Blot and Gel Data Policy*

Sincerely,

Ines

--

Ines Alvarez-Garcia, PhD

Senior Editor

PLOS Biology

Carlyle House, Carlyle Road

Cambridge, CB4 3DN

+44 1223–442810

Reviewers’ comments

Rev. 1:

In the manuscript by Lamire et al the authors explore the role of cytoplasmic pressure in morphogenesis of the Drosophila follicle. Specifically, how mechanical changes in nurse cells (NCs) lead to the flattening of the epithelial layer surrounding the follicle and the development of "stretched cells" (Stcs), a squamous epithelium, in the anterior part of the follicle. This process occurs along with the elongation of the follicle in the anterior-posterior direction (AP). Previous work from other labs had identified a role for the basal membrane (BM), laid down in parallel rings as the whole follicle rotates along the AP axis. While it has been shown that the BM is required for the development of ellipsoid follicles, it is unclear how the shape is generated, leading researchers to hypothesize that internal pressure generated by the expansion of NCs acts together with the confinement of the BM to produce the characteristic ellipsoid shape of the follicle.

In this manuscript Lamire et al used atomic force microscopy (AFM), genetic manipulations and light imaging to probe the system. Using a limited set of genetic manipulations, the authors firt concluded that the asymmetry in the follicle arises from mechanical features. Then, using AFM they were able to detect differences in stiffness along the AP axis that they were able to correlate with the differential flattening of Stcs in the anterior follicle as well as with the posteriorly orientation of cell membrane curvature in NCs. They then used experimental values of follicle curvature and stiffness to calculate the internal pressure of NCs, assuming a direct relationship and no interference from the BM or the Stcs, to conclude that a gradient of internal pressure exists in NCs and that this gradient plays an important role in the asymmetrical morphogenesis of the follicular epithelium. They then tested their model by the performing AFM measurements and morphological analyses together with genetic manipulations that alter key aspects of follicle development, and also correlate their observations with insights into ring canal characteristics.

Overall, this is an interesting, thought provoking and well written manuscript that is built around a logical and plausible scenario that reaches interesting conclusions that may be of broader significance. The conclusions are very interesting and likely correct at a basic level, i.e. that the pressure created by NCs is critical for the morphogenesis of the follicle. However, the authors used a technique (AFM) that provided a minimally invasive, yet indirect method and they also made a number of assumptions that, while reasonable, were not sufficiently tested and validated experimentally. Particularly, they assumed that the AFM data reflects the properties of NCs while the elaborate BM and the Stcs are in between the probe and the NCs.

To better support their approximation and their conclusions the authors should:

1-Perform a number of physiological manipulations including hypo and hyperosmotic media (using various osmolites, not just 1 M NaCl addition). 1a-Do different NCs respond to osmotic manipulations differently?

2-Test how changes in the extracellular matrix affect AFM measurements and possibly the pressure inference.

3-Test more extensively the contribution of the cell cortex and the follicular epithelium to the AFM data.

Other points:

4-Given that the curvature of the follicle depends on the overall geometry of the tissue, which depends on the BM and other features, can the curvature of the follicle really be used to infer the pressure of in NCs? In other words, is this really a local value the way it is determined? Here a more elaborate demonstration and theoretical validation is needed.

5-Could the posterior orientation of membrane curvature in NCs result simply from developmental timing? i.e. the anterior NCs expand first and so on.

6-How does the initial elliptical geometry impact how the internal pressure acts on the structure? Perhaps some discussion point could be added.

7-Hydrostatic pressure is isotropic, how can there be a gradient of pressure with an enclosed structure? Here, a more elaborate concept needs to be introduced to account for the proposed model. 7b-Similarly, what prevents the NCs with higher pressure to expand at the expense of the other cells? Are ring canal features sufficient?

8-What is the origin of NC pressure? Are there channels that could be manipulated?

Minor points:

9-Can the authors exclude that the variations in curvature they observe in Figure 1-H-J are not due to uneven digestion of the matrix?

10-Do different NCs respond differently to osmotic treatments under the conditions used in Figure 1 (i.e. collagenase treatment).

Very minor points:

-the authors use "Pression" in figures, do they mean "pressure"

-line 231: should be "compared"

-Reference 78 should be "Adams, Keller and Koehl". This is a truly insightful and seminal paper that should be cited here, but it did not show that pressure leads to axis elongation, that was actually done by Ellis et al (PMID 23460678).

Rev. 2:

This is an extremely original study and a fascinating question. This work addresses the complex question of how the mechanical properties of one group of cells influences the properties of neighboring cells, and how this interaction shapes the final organ these cells are part of. This work is not without conceptual and technical difficulties, intrinsic to the complex question being addressed, and I think the authors do a great experimental work.

Major points:

1. Intrinsic to a complex question being studied, I think one has to be careful on the conclusions, and this is one of the main problems with the manuscript. Often the authors conclude a causal relationship when they should only conclude that there is an important correlation, which may be causal or not. There are a few examples of this problem, where I think the authors over-interpret results and make statements that are not always fully supported by the data. For example, the order of events is difficult to prove: e.g. line 405 says that 'These together (NC pressure and FC shape changes) sculpt the final organ, the egg, by indirectly acting on the mechanical properties of the basement membrane'. However, the authors have not shown this to be the case. There is no data demonstrating that changes in the BM are due to the NC pressure and the FCs flattening…there is a correlation but not a causal effect. Another example: Fig1H-I: Could the collagenase differential effect be related to differences in BM composition/sensitivity to treatment? And not due to differential NC cytoplasmic pressure?

Also, Line 302: Title 'High cytoplasmic pressure, and subsequent StC flattening, leads to enhanced anterior follicle expansion'… this title is misleading...the experiments here show that flattening and/or BM softening contribute to anterior expansion.

2. The authors use various original and valuable techniques to measure stiffness and pressure of the tissue/cells. One of these techniques is AFM. This approach is valuable, and the results obtained I believe are robust, but I fail to see how the AFM values are directly linked to only NC cytoplasmic pressure to the point that the authors say 'demonstrates the presence of a cytoplasmic pressure gradient …" line 209…aren't the BM and the FCs also contributing to any values obtained with AFM? At this point the authors cannot say 'demonstrate' and only when put together with other results, then they could say there is strong data suggesting…

3. Another tricky part, which I guess is a limitation of the system or the approach, is the fact that the gradient of pressure should start at st8-early9, for it to be functional connected with flattening of FCs. But most measurements are later than that. Could the authors explain why more measurements are not taking place at st8, early st9, and most data presented is st9-st10? What are the limitations for measuring the pressure at earlier stages?

4. The last point that I found hard to get my head around, and that I think affects all experiments, is the staging of the follicles. How was staging exactly defined? The authors say various parameters were used to define the stage of the follicles, but how exactly? I think the authors need to be much more precise on how each stage is quantified, for each mutant and for each experiment.

For example, line 273: 'quantification of StC apical surface area shows that dic mutant follicles start flattening later than WT…' how do you know this? The precise staging of both the wt and mutant follicles is essential to draw such a conclusion. This is just one example.

In a similar manner, how do the authors know what is row1, 2, 3 etc...when the epithelia is disorganized, as it is the case in some mutants.

Other points:

- When using AFM on the anterior NCs…how far is the probe from the anterior tip? The authors rightly mentioned that it is difficult to access the tip, so how far? And how consistent?

- Reduced Myosin2 has no phenotype on progressive flattening, but treatment by the rock inhibitor shows defects….explain. Also, regarding the possible role of actomyosin activity, I feel that the authors have not explored enough genetic options. There are several RNAis, DN and overexpression constructs that they could use for manipulating the activity of zipper and sqh.

- dic was already used as a mutant to study FCs flattening, where it was shown for the first time that there is a link between NC growth and FC flattening. I think the authors do not introduce this early enough, and only present it briefly in one sentence after their own data (line 278: It also confims that NC growth impacts the degree of StC flattening as previously mentioned [32]…) I think this undermines the previous work.

- Fig2. A. There is a gradient of tension from mid-to-late st9 between anterior and C, but not with P.

B. calculated how in this fig? AFM measurements? The authors say AFM measurements, but as mentioned above, this is probably not only reflecting NC cytoplasmic pressure. Similarly, in C) Schematic representation of mean cytoplasmic pressure in each region of WT S10…' not necessarily only of cytoplasmic pressure, but of AFM results, which probably include BM and FCs properties as well.

H: what are the values 0-100? % of cells with the type of curvature represented by colors?

- Fig3. Can you show membrane curvature of dic and kel mutants at the same stage than WT, and that being at early st9 for all cases, when the data is more relevant for the flattening process? Fig3 shows wt st10, and late st9 dic.

Fig3 title implies that the gradient controls flattening, but I fail to see this in the results presented in this fig. The stretching of dic and kel cells seems alike in fig H' and I', and both seem to have higher area than controls… in detail, Fig3k: The quantification of area says that dic has lower area than wt in arrows 1-5. I fail to see this in the representing image of dic on fig3H and H', where cells 2, 3 and 4 seem to have higher area than wt in G and G'.

- line 273. Here the authors mention that there is no gradient of flattening, but there is still a small gradient, although reduced.

Also in this section, line 274: 'In contrast, kel..' I partially disagree with this statement, as the ratio of kel mutant looks like dic mutant, with less pronounced gradient in the C region

- Fig4. Could the authors point out in the images in A-C what area is Ant, Central and Post here? it would be nice to see an staining of the stages and ROI used for panel D. And, as mentioned above, this is one case where it is difficult to understand how the stages were defined here.

fig4e: can the authors show a pMad staining of a st7 wt to show that there is no pMad expression as stated in the text?

Also the quantification in fig4I shows no major differences in levels between the dic and wt of the same length, stats?

- Fig5. A' image has been heavily cropped around the egg chamber, why is that?. In B. what's the color scale representing? velocities? In H-H''. the StC seem also broken? can the authors show an overlapping to see how the cells and the membrane breaks relate to each other?

- Fig6J. One of the important findings here could be the difference between A, C and P (at each stage), but these stats are not calculated. The calculated stats are only for RCs at the same area, at different stages.

- figS6H, the kel follicle is showing a weird oocyte. What is that? Explain

Other points 2:

- Line 97 revise, something is missing

- Line 133 revise, something is missing

- TkvA: is this CA working? Shouldn't you see at least an effect on the flattening timing, even if the progression is normal?

- Fig1. '(B) Adherens junctions (Ecad) remodelling …. Both parameters are marker of flattening (see movie 1)….' where is this movie 1 on flattening parameters? The movie 1 I saw does not cover this at all.

- Line 429 talks about micropipette aspiration, but I failed to see these results

Rev. 3:

This is an extremely interesting paper on the mechanics of development in Drosophila, focusing on follicle development in Drosophila. The authors performed extensive experimentation, including AFM-based measurements of pressure in nurse cells, and demonstrated very clearly that there

exists a well-defined gradient in pressure from anterior to posterior that is closely linked with morphological changes in cells. The logic of the paper appears sound to me, and I think the work will be of great interest to developmental biologists of many kinds.

The only points I would like to make is that the authors have consistently used the Laplace pressure Delta P = 2 gamma/R to deduce the pressure from measurements of cell wall radii of curvature R. First, the factor 2/R only holds for spherical interfaces (where the sum of the two principal curvatures is just 1/R+1/R=2/R). For a general interface it is the mean curvature that matters. Thus, I would like to know the precise 3D geometry of the interfaces that are seen as 2D projections in e.g. Figure 1H. Are these actually sections of spheres? Second, and more importantly, I would have thought that the authors would use their measurements of pressure and curvature to deduce the typical values of the surface tension. If I put in numbers, with Delta P ~ 150 Pa and R ~ 25 microns (2.5x10^-5 m), I find gamma ~ 2x10^-3 N/m ~ 2 dyn/cm~2 erg/cm^2, which for comparison is about 1/40th of the surface energy of water against air. Is this a sensible number? What does it tell us about the cells in this system? How does it compare with other systems?

---

## [Decision Letter · Decision Letter 2]

17 Sep 2020

Dear Dr Grammont,

Thank you for submitting your revised Research Article entitled "Gradient in cytoplasmic pressure in the germline cells controls overlying epithelial cell morphogenesis" for publication in PLOS Biology. I have now obtained advice from the three original reviewers and have discussed their comments with the Academic Editor. 

We're delighted to let you know that we're now editorially satisfied with your manuscript. The only request we have is a slight edit in the title: "Gradient in cytoplasmic pressure in germline cells controls overlying epithelial cell morphogenesis." The reviewers' comments are attached below.

Before we can formally accept your paper and consider it "in press", we also need to ensure that your article conforms to our guidelines. A member of our team will be in touch shortly with a set of requests. As we can't proceed until these requirements are met, your swift response will help prevent delays to publication. Please also make sure to address the data and other policy-related requests noted at the end of this email.

- a cover letter that should detail your responses to any editorial requests, if applicable

*Copyediting*

*Published Peer Review History*

*Early Version*

Sincerely,

Ines

--

Ines Alvarez-Garcia, PhD,

Senior Editor,

ialvarez-garcia@plos.org,

PLOS Biology

Thank you for sending us some of the data we require in our Data Policy, however we are still missing the data underlying the graphs represented in the following supplementary files:

Fig. S2A, B, D, F, G, H; Fig. S3B, C; Fig. S4C, E, F, G, H, I; Fig. S6A, E, G, H; Fig. S7J, K and Fig. S8A, B, C, D, E

Please also ensure that figure legends in your manuscript include information on WHERE THE UNDERLYING DATA CAN BE FOUND.

Reviewers’ comments

Rev. 1:

I think the authors addressed the issues I had raised, and think also those from other reviewers, satisfactorily and in my opinion the manuscript is now acceptable for publication.

Rev. 2:

Accept.

Rev. 3:

The authors have done an excellent job replying not only to my questions but the very extensive ones of the other reviewers. The paper should definitely be published in its present form.

---

## [Editor Report · Decision Letter 3]

13 Oct 2020

Dear Dr grammont,

On behalf of my colleagues and the Academic Editor, Nicolas Tapon, I am pleased to inform you that we will be delighted to publish your Research Article in PLOS Biology. 

Early Version

PRESS 

Kind regards,

Alice Musson

Publishing Editor, 

PLOS Biology

on behalf of

Ines Alvarez-Garcia,

Senior Editor

PLOS Biology